# Listening to the Wise Few: Query–Key Alignment Unlocks Latent Correct Answers in Large Language Models

## Abstract

Language models often struggle with multiple-choice question answering (MCQA) tasks when they fail to consistently choose the correct letter corresponding to the right answer. We find that while certain attention heads within large language models (LLMs) identify the right answer internally, this information can be lost before the final decision stage of what to output. To demonstrate and measure this effect, we introduce the Query-Key (QK) score, a metric based on query- and key-vectors alignment, that retrieves the correct answer directly from individual attention heads. This allows us to identify "select-and-copy" heads, that consistently focus on the correct option during inference. Across four standard MCQA benchmarks (MMLU, CosmosQA, HellaSwag, and HaluDialogue), QK score from such heads can be better than the model's own output by up to 16% in terms of accuracy, especially for smaller models. On a synthetic dataset, these heads may outperform the baseline by as much as 60%. We also find that QK-scores from "select-and-copy" heads are robust to option permutations and remain effective in few-shot settings. After analyzing a wide range of models across the LLaMA, Qwen, and other families, with 1.5B to 70B parameters, we observe the select-and-copy phenomenon in all of them. Our findings offer new insights into the inner workings of LLMs and open a principled path toward head-level interventions for controllable and trustworthy LLM reasoning.

## 1 Introduction

Questions with multiple answer options are a common form of benchmark for evaluating question answering (Hendrycks et al., 2021), common sense reasoning (Zellers et al., 2019), reading comprehension (Huang et al., 2019), and other capabilities of Large Language Models (LLMs). In multiple choice question answering (MCQA) tasks, the model is provided with the question and a set of answer options, e.g. *"Question: How many natural satellites does the Earth have? Options: A. 0. B. 1. C. 2. D. 3."* Some context may also be provided to help answer the question. The model is then expected to select the letter corresponding to the correct answer. This format resembles real-life student exams and enables straightforward evaluation using automated tools.

However, this format introduces challenges that go beyond raw accuracy. LLMs are expected not only to reason correctly, but also to comply with a rigid output format, which is not always trivial, especially for smaller models. Some models may provide correct answers with formatting issues or additional commentary before the option label, complicating automatic evaluation. As a result, some studies delegate answer evaluation to another LLM instead of relying on exact string comparison (Wang et al., 2024).

Further complicating the evaluation, LLMs can sometimes follow shallow patterns, such as favouring certain answer options (e.g., always choosing "A" or "D"), which can skew the results (Zheng et al., 2024a). These issues raise deeper questions about what the model actually knows versus what it outputs and highlight the need for more interpretable and robust evaluation methods.

In this work, we take a white-box interpretability approach to MCQA. Rather than relying solely on the final output, we analyse the internal attention patterns of LLMs during inference to uncover how they internally represent and process the answer options. Our key insight is that some attention

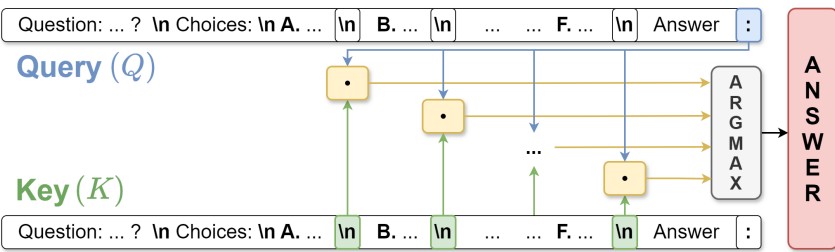

Figure 1: Our method calculates the Query-Key score between the end-of-line token of an answer option and the last token of the prompt for the designated head, from which we derive the answer.

heads perform a *select-and-copy* operation: they identify semantically relevant options and propagate their embeddings forward in the network. Building on this, we introduce a novel method that uses the query-key interactions of these heads to derive the correct answer without having to parse the model's output string.

When answering multiple-choice questions, LLMs internally compute rich semantic representations of both the question and the answer options. We observe that specific heads align these representations through the query-key attention mechanism and guide the selection of the correct option often more reliably than the model's final output. By identifying and analyzing these select-and-copy heads, we uncover interpretable mechanisms underlying MCQA performance. Our method exposes consistent patterns across models of various sizes (1.5B to 70B parameters), revealing that middle layers are particularly effective at encoding task-relevant information. Later layers, in contrast, tend to override this information, sometimes degrading performance. Notably, the best-performing heads are consistent across datasets and task variations, and the answers generated by these heads are often more accurate than the model's final output, particularly in zero-shot scenarios.

Our contributions are as follows: (1) We show that select-and-copy heads are present in LLMs ranging from 1.5B to 70B parameters, performing the option selection operation for MCQA tasks, and the best of them are universal and consistent across datasets; (2) We introduce the Query-Key (QK) Score, a novel option-scoring method based on key-query representations from these heads. Using it help to improve accuracy by 9-16%, and also provides information about which model components contribute most to MCQA task solving; (3) We demonstrate that our method is more stable than baselines when handling option permutations or the addition of supplementary options (e.g., "I don't know"); (4) Our results support the hypothesis that semantic representations are encoded in specific heads in the query, key, and value vectors of a phrase's final tokens (namely, end-of-sentence or end-of-line tokens), as observed in previous studies (e.g., Li et al. (2023b); Stolfo et al. (2023)); (5) We analyse the attention patterns of *select-and-copy* heads and their behaviour under different conditions, advancing our understanding of how attention mechanism can work as "select and copy" operation, and how LLMs function in general.

## 2 RELATED WORK

Question-answering tasks are commonly used to evaluate Large Language Models (LLMs) in knowledge retention, text comprehension, and reasoning abilities. The results of such an evaluation are presented in technical reports on recent LLMs, Touvron et al. (2023); Dubey et al. (2024); OpenAI (2024); Anthropic (2024). Multiple-choice question answering (MCQA) is a common benchmark due to its straightforward evaluation format (Ye et al., 2024; Pal et al., 2022). MCQA is typically approached using multiple-choice prompting (MCP), where all answer options are presented at once, or cloze prompting (CP), where options are evaluated individually. MCP, despite biases in answer order and selection (Gupta et al., 2024; Pezeshkpour & Hruschka, 2024; Zheng et al., 2024a), offers several advantages over cloze prompting. In MCP, the model sees all the options and can compare them and also we do not need to normalise the answer probabilities as in CP (Robinson & Wingate, 2023).

In this work, we investigate the inner mechanisms of LLMs, particularly the role of attention heads in MCQA tasks. Attention head functions have been studied since early transformer architectures (Jo &

Myaeng, 2020; Pande et al., 2021), and are now a core focus in mechanistic interpretability for decoder-only models (Elhage et al., 2021; Olsson et al., 2022; Bricken et al., 2023). For example, *induction heads*, identified by Elhage et al. (2021), are crucial for in-context learning (Olsson et al., 2022; Von Oswald et al., 2023), indirect object identification (Wang et al., 2023), and overthinking (Halawi et al., 2024). Other task-specific heads include constant heads (Lieberum et al., 2023), negative heads (Yu et al., 2024), and content gatherer heads (Merullo et al., 2024). For further details on mechanistic interpretability, see Rai et al. (2024) and Zheng et al. (2024b).

We focus on *select-and-copy heads*, which identify the correct option in MCQA task. We analyse different types of option-representative tokens that these heads may attend to. Our experiments reveal that the most effective heads for MCQA are in the middle layers of the LLM. This aligns with prior research showing that while early layers encode substantial information, it is often lost or altered in later layers (Kadavath et al., 2022; Azaria & Mitchell, 2023; Liu et al., 2023; Zou et al., 2023; Lieberum et al., 2023; CH-Wang et al., 2024). While previous studies primarily use linear probes on hidden representations (Ettinger et al., 2016; Conneau et al., 2018; Burns et al., 2023), we show that discrepancies between model output and internal structures can also be captured through query-key interactions.

## 3 ATTENTION AS SELECT-AND-COPY ALGORITHM

In this section, we describe how the attention mechanism can function as a *select-and-copy* operation. Consider a sequence of $N$ token embeddings, $\{\boldsymbol{x}_i\}_{i=1}^N$, each attention head performs a transformation of the input embeddings:

$$\boldsymbol{o}_m = \sum_{n \leq m} a_{m,n} \boldsymbol{v}_n, \quad a_{m,n} = \frac{\exp\left(\frac{\boldsymbol{q}_m^\top \boldsymbol{k}_n}{\sqrt{d}}\right)}{\sum_{j=1}^N \exp\left(\frac{\boldsymbol{q}_m^\top \boldsymbol{k}_j}{\sqrt{d}}\right)}, \tag{1}$$

where $\boldsymbol{q}_i = \boldsymbol{W}_q \boldsymbol{x}_i$, $\boldsymbol{k}_i = \boldsymbol{W}_k \boldsymbol{x}_i$, $\boldsymbol{v}_i = \boldsymbol{W}_v \boldsymbol{x}_i$ and $\boldsymbol{W}_q, \boldsymbol{W}_k, \boldsymbol{W}_v \in \mathbb{R}^{d_{model} \times d}$ are learned weight matrices.

This means that the $m$-th token of the output embedding is a linear combination of the values of the preceding tokens, weighted by the $m$-th row of the attention matrix $\boldsymbol{A} = \{a_{n,m}\}_{n,m=1}^N$. If all but one component of this combination are close to zero, the transformation can be interpreted as a conditional copy mechanism. Specifically, if $a_{m,j}$ is the only non-zero weight in the $m$-th row, then, due to stochastic nature of $\boldsymbol{A}$, $a_{m,j} \approx 1$, and $\boldsymbol{o}_m \approx \boldsymbol{v}_j$. Each token position from 0 to $m$ can be viewed as a cell storing the corresponding value vector, with the attention weights $a_{m,i}$ determining the *choice* of which cell to copy to the $m$-th output.

We introduce the concept of *select-and-copy* heads that select the appropriate option label and copy it to the output, based on the semantic content of the options, rather than their positional order. To extract information from heads, we use query vectors and key vectors from them. However, in modern models these vectors usually undergo additional transformation with positional encoding, such as Rotary Position Embedding (RoPE) (Su et al., 2024). To mitigate the potential impact of the relative position shift, we introduce the *Query-Key-score (QK-score)*, which does not incorporate positional shifts when comparing queries and keys (see details in the next section). We compare this score to the *Attention-score*, which is computed after the RoPE is applied.

## 4 APPROACH

Consider an MCQA task with the corresponding dataset $\mathcal{D} = \mathcal{D}_{val} \cup \mathcal{D}_{test}$, where each instance consists of a prompt, a question, and labeled answer options. Given this input, the model is tasked with generating the label for the best answer option.

We investigate two ways to identify the heads in the model that implement the option selection mechanism described above. The first one is the following: we select the heads that perform best using $\mathcal{D}_{val}$, based on their accuracy on this validation set. We then evaluate their performance on the much larger $\mathcal{D}_{test}$. If these heads are indeed responsible for option selection, their performance should be at least comparable to that of the entire model. We confirm this claim through experiments presented in Section 6. The second method for selecting such heads is proposed in Section 7, where

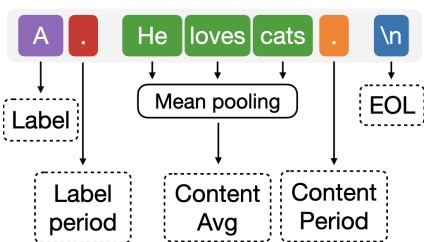 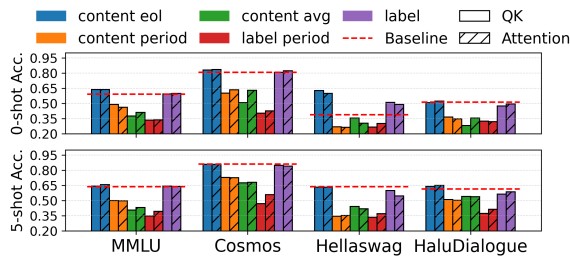

Figure 2: (Left) Scheme for option-representative token types. (Right) Performance of *QK-score* and *Attention-score* (App. E) for different option-representative tokens on LLaMA3.1-8B base.

we analyse of attention maps and demonstrate that the best performing heads, chosen that way, also effectively implement the option selection algorithm described above.

**QK-score and Attention-Score.** Given a data sample for an MCQA task, we denote by $q$ the question, which may be supported by context if applicable, by $o = \{o_1, o_2, ..., o_n\}$ the semantic content of the provided answer options, and by $d = \{d_1, d_2, ..., d_n\}$ the corresponding labels (e.g., A/B/C/D). We assume that the labels are ordered by default, see Figure 1 and Appendix A.2 for examples. The model is tasked with estimating $P(d_i \mid q, d, o)$ – the probability of selecting option $d_i$ given the question $q$, the options contents $o$, and the options labels $d$.

Let $t_i$, where $i \in \{1, 2, \ldots, n\}$, represent the indices of tokens that encode information about the corresponding answer options. We refer to these as *option-representative tokens*. Given the causal nature of the attention mechanism in LLMs, the most logical choice is the last token following the option's content—typically the "\n" token. Our analysis of different tokens across all models supports this claim (Figure 2), therefore we use the end-of-line token following the $i$-th option content as $t_i$ in most of our experiments. However, we also found that, in some models, the label token might serve as an option-representative token, although this behaviour is not consistent across models. We discuss this behaviour and results for other models in Appendix B.

Let $N$ denote the length of the entire text sequence, and consider the attention head with index $h$ from layer $l$. Given the triplet $(q, d, o)$, we define the *QK-score* $S_{QK}^{(l,h)}(d_i)$ for the option $d_i$ as the dot product of the query vector of the whole sequence and $t_i$-th key vector:

$$S_{QK}^{(l,h)}(d_i) = \boldsymbol{q}_N^{(l,h)\top} \boldsymbol{k}_{t_i}^{(l,h)} \quad i \in \{1, 2, ..., n\} \tag{2}$$

We also define an auxiliary Attention score $S_{Att}^{(l,h)}(d_i) = a_{N,t_i}^{(l,h)}$ (see Appendix E).

The prediction is straightforward: we select the option with the highest score. As we do not apply a positional transformation in the QK score, it does not correspond to the attention scores prior to the Softmax operation in Transformer block. As a result, the token with the highest QK-score does not necessarily correspond to the token with the maximum attention score. For an example, see Figure 52.

## 5 EXPERIMENTS

### 5.1 DATASETS

At first, we introduce **Simple Synthetic Dataset** (SSD), a synthetic task in the MCQA setting designed to evaluate the model's ability to handle the basic task format. Tasks in the SSD do not require any factual knowledge from the model. The main version of this dataset consists of questions in the form, "Which of the following options corresponds to "<word>"?" and contains 2.500 examples. The options consist of a word from the question and three random words, all mixed in a random order and labeled with the letters 'A'-'D'. Other variations of this dataset contain smaller versions in different languages (see Appendix I), as well as versions with a different number of options and different data labels, all sampled and named according to the same principle (see Appendix H for more details).

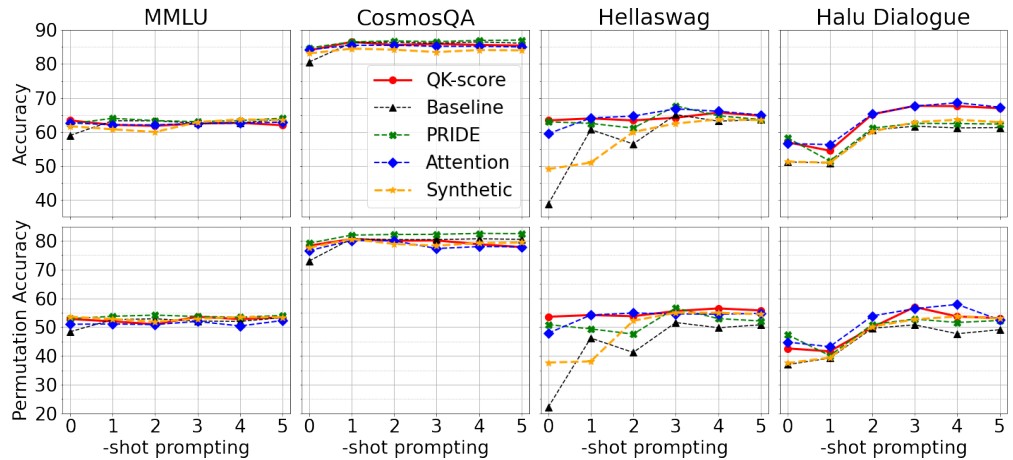

Figure 3: Comparison of different methods for LLaMA3.1-8B. Reported metrics are Accuracy and Permutation Accuracy.

Besides, we conduct experiments on four challenging real-world MCQA datasets from LLM benchmarks: **MMLU** (Hendrycks et al., 2021), **CosmosQA** (Huang et al., 2019), **HellaSwag** (Zellers et al., 2019) and **HaluDialogue**, which is a "dialogue" part of HaluEval (Li et al., 2023a). From each dataset we picked 10,000 questions with four possible answer options, and some samples also include a context that must be used to determine the correct answer. More details about each dataset and examples can be found in Appendix A. Our prompt has no instructions, only provides necessary information for the task, and we ensured the performance was stable under the change of prompts, see Appendix D.

Finally, following Ye et al. (2024) in all five datasets we specially modified questions by adding two extra options "*E. None of the above.*" and "*F. I don't know.*" that are intended to aggregate the uncertainty of LLM. Despite adding these two options, there are *no* questions for which 'E' or 'F' are correct answers. We provide more intuition about adding these options in Appendix C.

## 5.2 BASELINES

The standard approach for MCQA is to use output probabilities from LLM on options $d_i$ to pick the final option $\hat{d}$:

$$\hat{d} = \arg\max_{d_i} P(d_i \mid q, d, o), \tag{3}$$

where $q$ is the question, $o = \{o_1, \ldots, o_n\}$ – option contents, $d = \{d_1, \ldots, d_n\}$ – options labels. In our experiments, we refer to this method as BASELINE.

In Zheng et al. (2024a), it was proposed to mitigate option selection bias by averaging the results over option permutations. The idea is to use the set of all cyclic permutations $\mathcal{I} = \{(i, i+1, ..., n, 1, ..., i-1)\}_{i=1}^{n}$ to calculate the debiased probability:

$$\tilde{P}(d_i \mid q, d, o) = \frac{1}{|\mathcal{I}|} \sum_{I \in \mathcal{I}} \log P(\pi_I(d_i) \mid q, d, \pi_I(o)) \tag{4}$$

Since computing probabilities for all permutations for each question is computationally expensive, authors propose to estimate the prior distribution for option labels on validation set which contains $5\%$ of all samples, and use it to debias new samples. In our experiments we refer to this method as PRIDE. The test set is the same as the one that we use for head selection. We employ this method to evaluate the QK-score's effectiveness in addressing selection bias and to compare it with the model's performance after debiasing.

Following previous studies on selection bias Pezeshkpour & Hruschka (2024); Zheng et al. (2024a), we test whether our methods tend to choose specific option rather than the correct answer. In our experiments, we observed that both the *QK-score* and the *Attention-score* may occasionally exhibit

bias toward certain answer options, partially reflecting the overall model bias (see Appendix L). This suggests that debiasing methods could potentially improve performance. However, investigating such approaches is left for future work.

### 5.3 EXPERIMENTAL SETUP

Our main experiments were carried out the following way: first, we took a pretrained LLM and froze its parameters. Following Zheng et al. (2024a), we selected best head in terms of QK-score on $\mathcal{D}_{val}$, which is a fixed set of $5\%$ samples from $\mathcal{D}$ for each dataset. Then we calculated QK-Scores and Baseline on samples from $\mathcal{D}_{test}$ only with this head. If several heads had equal accuracy scores, we chose one from the lower level of the model (although this barely occurred in our experiments). Finally, we performed random shuffle of option contents in all questions and repeated the above procedure; this was done to correctly compute the Permutation Accuracy Gupta et al. (2024) metric. Option labels remain on place during shuffle and options "*E. None of the above.*" and "*F. I don't know.*" are exempt from shuffling. Also, note that it may be two different heads that achieve the best QK-scores in the validation set before and after the option permutation.

We report two quality metrics on the test subset: the accuracy of predicted answers (from the first run) and the Permutation Accuracy (PA). The latter is, in a sense, an accuracy, stable to option permutation. It is computed as the percentage of questions for which the model selects the correct choice both before and after the random permutation of options:

$$\text{PA} \;=\; \frac{1}{N}\sum_{i=1}^{N}\text{I}_i\text{I}_i^p,\tag{5}$$

where $\text{I}_i$ is the indicator value equals to 1 iff the model answers question $i$ correctly, while $\text{I}_i^p$ equals to 1 iff the model answers question $i$ correctly after answer contents were permuted.

Also similar to Zheng et al. (2024a) we used $N$-shot setup, where we provide model with additional examples of MCQA task. The examples are separated from each other and from the actual question by single line breaks. Refer to Appendix A.3 for examples. The few-shot examples are the same for every question in the given dataset. The set of examples for $(k+1)$-shot extend the set of examples for the $k$-shot prompts, adding one new example. The few-shot examples were chosen from the first fifteen entries of the validation set. Their selection was mostly arbitrary, but we tried to filter out questions that we considered suboptimal from the perspective of an English-speaking human expert.

Even without a task-specific $\mathcal{D}_{val}$, we can identify predictive heads using our synthetic data. We first rank all heads by their QK-score on the synthetic set and select the top $5\%$. In an $N$-shot setup, we then remove any of these heads that answered all in-context examples incorrectly. The final prediction is the option with the highest average QK-score across the remaining heads. Figure 3 refers to this procedure as 'Synthetic'. A similar aggregation can be applied to heads chosen on a given $\mathcal{D}_{val}$ (see Appendix), though a comprehensive study of head combinations is beyond this work's scope.

All models in our evaluation were run under the same metrics, prompts, and experimental setup.

## 6 RESULTS

### 6.1 SYNTHETIC DATASET RESULTS

Our experiments on the synthetic dataset demonstrate the remarkable efficiency of the QK score. It achieves near-perfect accuracy across all tested models, languages, and option counts, while full LLMs degrade as the number of choices increases(Fig. 5, Table 1; more results in Appendix). Although individual attention heads exhibit diverse behaviors, each model contains a subset of "stable heads" whose accuracy remains consistent across languages and option sizes. Importantly, as we show further, some of these stable heads retain their performance on real data.

Table 1: Top-20 heads performance on Simple Synthetic Dataset for LLaMA 3.1-8B

| Language | Max Acc | Min Acc |
|----------|---------|---------|
| English | 1.000 | 0.995 |
| Italian | 1.000 | 0.995 |
| French | 1.000 | 0.985 |
| Russian | 1.000 | 0.990 |

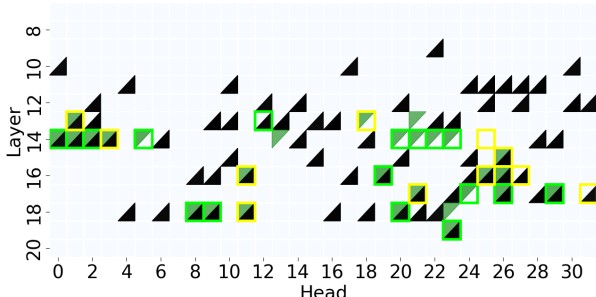

Figure 4: Heatmap of best-performing heads in LLaMA3.1-8B. Colored frames mark heads that generalize the best across real datasets (green: top 5% across all four datasets, yellow: across three of them). Dark triangles show heads beating the baseline on synthetic data; green triangles - on real data. Layers 0–7 and 21–32 are omitted due to lack of notable heads.

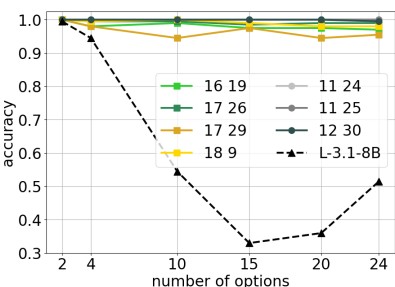

Figure 5: QK-score accuracy for SSD with varying numbers of options in a zero-shot setting for LLaMA-3.1-8B. Square markers indicate heads performing well on real datasets (see Fig. 4); round markers highlight heads good specifically on synthetic data; triangle-dotted line shows the baseline.

| Method | | LLaMA... | | | | LLaMA... (chat, instruct) | | | |
|---|---|---|---|---|---|---|---|---|---|
| | | 3-8B | 3-70B | 3.1-8B | 3.1-70B | 3-8B | 3-70B | 3.1-8B | 3.1-70B |
| | | **MMLU** | | | | | | | |
| Baseline | Acc | 60.3 | **75.3** | 59.1 | 72.9 | 60.5 | **78.2** | 60.4 | 73.8 |
| | PA | 50.4 | **68.8** | 48.6 | 65.7 | 47.7 | **70.1** | 50.1 | 67.8 |
| QK-score | Acc | **61.0** | 74.5 | **62.5** | **75.9** | **63.0** | 77.9 | **64.9** | **79.4** |
| | PA | **51.5** | 66.0 | **50.7** | **68.5** | 49.3 | 67.9 | **55.2** | **74.4** |
| | | **Cosmos QA** | | | | | | | |
| Baseline | Acc | 54.9 | 82.0 | 80.7 | 88.9 | 85.4 | 91.6 | 87.2 | 83.4 |
| | PA | 39.3 | 75.7 | 73.1 | 85.7 | 71.0 | 82.5 | 82.7 | 79.6 |
| QK-score | Acc | **70.6** | **87.6** | **84.2** | **91.3** | **88.6** | **94.1** | **90.9** | **94.7** |
| | PA | **60.9** | **81.7** | **78.6** | **87.4** | **75.1** | **88.1** | **86.8** | **92.6** |
| | | **Hellaswag QA** | | | | | | | |
| Baseline | Acc | 33.5 | **82.5** | 39.2 | 69.5 | 67.4 | **86.8** | 68.4 | 74.7 |
| | PA | 15.8 | **76.1** | 22.5 | 59.6 | 27.8 | 71.2 | 58.7 | 68.8 |
| QK-score | Acc | **60.9** | 82.1 | **64.0** | **85.2** | **72.5** | 86.3 | **71.5** | **86.7** |
| | PA | **50.8** | 75.2 | **54.3** | **78.0** | 36.3 | **72.8** | 65.3 | **80.8** |
| | | **Halu Dialogue** | | | | | | | |
| Baseline | Acc | 46.6 | 44.3 | 51.5 | 18.9 | 62.1 | 68.8 | 55.8 | 62.9 |
| | PA | 29.1 | 33.5 | 36.6 | 11.8 | 42.6 | 63.8 | 47.4 | 62.1 |
| QK-score | Acc | **52.3** | **67.8** | **56.1** | **68.7** | **64.7** | **76.7** | **64.9** | **78.8** |
| | PA | **36.7** | **57.9** | **41.1** | **56.3** | **46.6** | **65.6** | **51.0** | **73.5** |

Table 2: Comparison of different base models in zero-shot setup on various Q&A datasets. Reported metrics are Accuracy (Acc) and Permutation Accuracy (PA). Best results are highlighted in **bold**. Detailed results for LLaMA3.1-8B are presented in Table 5.

## 6.2 REAL DATASET RESULTS

Figure 3 reports QK and attention-based scores for the LLaMA3.1-8B model, using heads selected on each validation set and the universal heads from synthetic data. In the zero-shot setting, our method outperforms the baseline by 5% on MMLU and up to 29% on HellaSwag. On knowledge-based tasks (MMLU, CosmosQA), this gap narrows as option counts increase, indicating that QK can recover latent answers when standard MCQA formatting hinders generation. Few-shot prompting can similarly guide the model by explicitly demonstrating the required MCQA format, although it incurs substantially higher computational cost. On common-sense benchmarks (HellaSwag, HaLuDialog), QK consistently surpasses the baseline for all option sizes, revealing knowledge suppressed during decoding.

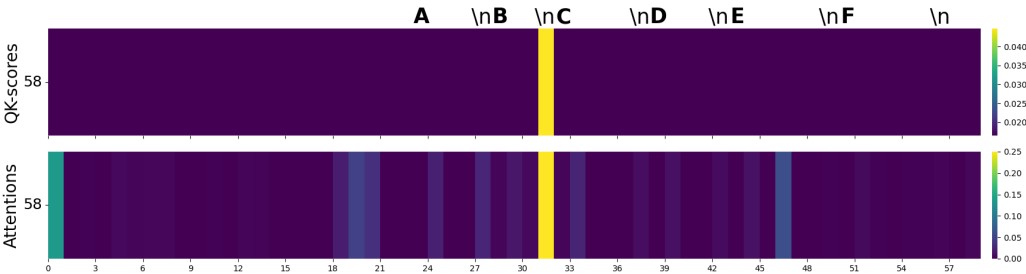

Figure 6: QK-scores after softmax and attentions for the last token on the 0-shot MMLU example on (17, 26) head of LLaMA3.1-8B. The task is "`Question:  What singer appeared in the 1992 baseball film 'A League of Their Own'\nOptions:\nA. Brandy.\nB. Madonna.\nC. Garth Brooks.\nD. Whitney Houston.\nE. I don't know.\nF. None of the above.\nAnswer:`" with correct option 'B'.

Comparing head-selection strategies, synthetic-data heads excel on the first two datasets but lag on the latter pair, whereas validation-selected heads deliver robust performance throughout. This suggests that tasks involving longer options, deeper reasoning, or discourse coherence require skills not captured by our simple synthetic task.

We observe similar trends for permutation accuracy. Applying PriDe bias-mitigation atop the baseline narrows the zero-shot gap and even marginally outperforms QK under few-shot prompting, but its performance is not stable across models and datasets. This highlights the importance of the information captured by select-and-copy heads and indicates that while debiasing contributes to improved outcomes, it addresses only part of the underlying issue.

Finally, Table 2 summarizes zero-shot results on larger LLaMa variants (2-13B, 2-70B, 3-8B, 3-70B) and their instruction-tuned counterparts; full few-shot evaluations appear in Appendix M. Appendix N reports additional results for Qwen 2.5 (1.5–72B) and smaller models (Gemma 2-2B, Dolly V2-3B, Phi-3.5-mini). Across most architectures and datasets, QK consistently unlocks hidden task-solving capacity that few-shot prompting only partially reveals, and identifies both universal and dataset-specific heads that reliably select correct answers.

**Key takeaways**

- All the tested models contain *select-and-copy* heads that encode a latent MCQA task-solving ability often exceeding what is reflected by the final generation.
- Few-shot prompting partially unlocks this intrinsic knowledge but requires additional examples and substantially more computation.

## 7 ANALYSIS

**Best heads.** Finding the single best head normally requires a validation set, so we ask whether a small "universal" subset can match dataset-specific heads. For each dataset and shot level, we take the top 5% of heads by QK-score (Appendix F) and count which appear most often. Figure 4 marks heads that are top on synthetic zero-shot (grey) and those ranking highest on real data from 0–4 shots (green). For LlaMa3.1-8B, most stable heads lie in layers 12–21. Four heads—(16,19), (17,26), (17,29), and (18,9)—consistently outperform the baseline on all real datasets, and ranked high on SSD. Notably, these heads exhibit cross-dataset robustness, achieving top performance across most of four real-world benchmarks without retraining or reselection. This suggests a degree of task-agnostic reliability in their answer-selection capability. For LLaMa2-7B, two analogous heads ((14,20) and (14,24)) emerge (see Appendix).

**Attention patterns analysis.** Figure 6 illustrates the attention map and QK-score for head (17,26); other examples appear in the Appendix. These *select-and-copy* heads strongly attend to the "\n" token following the correct option—exactly as expected (essentially, they select the end-of-span token with the highest semantic match to the query and inject its value vector into the residual stream. )—and the

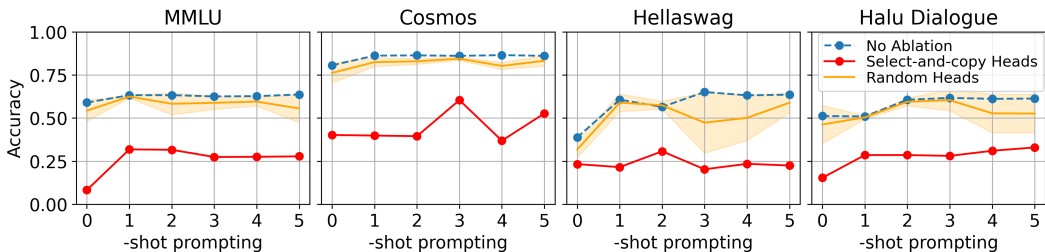

Figure 7: Zero-ablation of *select-and-copy* and random heads for LLaMA3.1-8B. The heads are selected by above-random performance of *QK-score*, and the random ablation with the same amount of random heads as selected heads is added for comparison. Random heads were ablated with 5 different launches, and we report mean and standard deviation of accuracy.

QK-score highlights this behavior even more clearly. Leveraging this, we propose an unsupervised head-scoring algorithm: heads with high total attention on option tokens and high variance across them tend to be stable. When applied to LLaMA3.1-8B, this score ranks head (16,19) near the top, but there are also high scorers which do not perform well (see Appendix K).

**Select-and-copy heads ablation.** Using zero-ablation (Olsson et al., 2022), we zero out selected attention heads and measure MCQA accuracy. Removing all select-and-copy heads drops performance to near random (Figure 7), confirming their causal role. Additional model results are in Appendix G.

**Key takeaways**

- A core set of select-and-copy heads is stable across languages, option counts, and tasks, consistently exceeding baseline performance.

- Although some heads show clear attention to option tokens, such patterns alone do not guarantee effective option selection.

- The select-and-copy mechanism is essential for accurate MCQA generation.

**Limitations.** Our method cannot be applied to models without an access to attention matrices. Also, our method is not applicable on scarce-resource tasks, even though one can utilize the heads we marked as robust enough.

Furthermore, our method is inherently tied to the structured nature of MCQA, where answer options are explicitly enumerated. In unstructured QA settings—e.g., free-form generation—the absence of discrete, aligned option tokens makes it unclear how to apply QK scoring directly. While the underlying "select-and-copy" mechanism may generalize to other retrieval-like tasks, extending our approach beyond MCQA remains an open challenge and promising direction for future work.

## 8 CONCLUSION

In this work, we introduce novel scoring mechanism – *Query–Key (QK) score*, derived from internal representations of LLMs. It provides new insights about how model solves MCQA task and offers an interpretable mechanism that raises MCQA accuracy by up to 16% across popular benchmarks and up to 60% on a synthetic dataset. It demonstrates that in these cases, the particular parts in the middle of the model already know the right answers to the questions, yet the model itself didn't learn to propagate this information to the last layer.

We catalogued a small, transferable subset of *select-and-copy* attention heads that consistently appear across model families, scales, and prompt formats and that remain effective under language switches or expanded option sets. Our findings suggest that these heads have the potential to deepen our understanding of LLMs' capabilities not only for MCQA but for other reasoning tasks as well. This work opens up new avenues for further research into the internal dynamics of LLMs, including a deeper exploration of attention mechanisms and their role in complex task-solving, or into head-level interventions for controllable and trustworthy LLM reasoning.

**Reproducibility Statement.** In this paper we use publicly available datasets; details and prompt templates are provided in Appendix A.1. The synthetic dataset and code notebooks containing method implementations and scripts for reproducing our experiments will be released along with the paper.

**Ethics Statement.** Our work is a foundational research on certain aspects of internal mechanisms within LLMs. There is no direct negative societal impact of our work that we can clearly identify.

In this research we employ assets (datasets and models) all of which are permitted for academic usage; we provide proper citations for them and ensure full compliance with their respective licences.

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

# A DATASETS

## A.1 DATASETS DETAILS

**Massive Multitask Language Understanding (MMLU)** (Hendrycks et al., 2021) contains 4-way questions on the variety of topics related to STEM, the humanities, the social sciences, and other fields of knowledge. We sample 10,000 instances from the test set to utilize them in our experiments. Available under the MIT license.

**CosmosQA**[1] (Huang et al., 2019) together with question and answer options additionally contains text paragraph that is supposed to be used by a model to give the final answer. The purpose is to evaluate the model's reading comprehension and commonsense reasoning capabilities. Similar to MMLU, we sampled 10,000 instances from the test set. Available under CC-BY-4.0 license.

**HellaSwag** (Zellers et al., 2019) evaluates the commonsense reasoning capabilities of the model by selecting the best sentence completion for a given sentence prompt, given a short text as a context. We also extracted 10,000 entities from this dataset. Available under the MIT license.

**HaluDialogue** is a "dialogue" part of HaluEval (Li et al., 2023a) dataset with about 10,000 examples. Here, a model is asked to choose an appropriate continuation of a dialogue from four possible options. Available under the MIT license.

We chose datasets in order to cover the main formats of questions and common NLP tasks. Since our primary intention was to focus on the investigation and interpretability of attention heads' roles in Question Answering, we limited ourselves to these four datasets. We did not try to cover as many benchmarks as possible.

## A.2 EXAMPLES OF QUESTIONS FROM DATASETS

Listing 1: MMLU example

```
Question: Where is the Louvre museum?
Options:
   A. Paris.
   B. Lyon.
   C. Geneva.
   D. Vichy.
   E. I don't know.
   F. None of the above.
```

Listing 2: CosmosQA example

```
Context: My house is constantly getting messy and I ca n't keep
    up . I am starting at a new school with no one I know and
    it is 4 times bigger than UAF . I am now going to have to
    balance school , homework , kids , bill paying ,
    appointment making and cleaning when I can barely keep up
    without the school and homework ( keep in mind this is a
    full time GRADUATE program at a fairly prestigious school )
     . We are in financial crisis .
Question: What is causing the narrator 's recent stress ?
Options:
   A. They are moving to a new house .
   B. I would have tried to guess their password and
       alternatively gone to a coffee shop for wifi.
   C. They are moving to a new university .
   D. They are moving to a new house for the kids .
   E. I don't know.
   F. None of the above.
```

Listing 3: HellaSwag example

```
Context: A young boy is wearing a bandana and mowing a large
    yard. he
Question: Which of the following is the best ending to the
    given context?
Options:
   A. is unrelieved by the weeds and is barely smiling.
   B. walks away from the camera as he pushes the mower.
   C. moves and walks the mower but gets stuck because he is
       engaged in a game of ping pong with another boy.
```

---

[1]https://wilburone.github.io/cosmos/

```
D. seems to be doing a whole lot of things and talks to the
   camera from behind a white fence.
E. I don't know.
F. None of the above.
```

Listing 4: Halu Dialogue example

```
Context: [Human]: I like Pulp Fiction. What do you think about
   it? [Assistant]: I love it. It was written by Roger Avary [
   Human]: I heard he also wrote The Rules of Attraction. Do
   you know who is in that movie?
Question: Which of the following responses is the most suitable
    one for the given dialogue?
Options:
   A. Swoosie Kurtz is in it.
   B. Fred Savage is in it.
   C. Yes, it is a drama and crime fiction as well. Do you like
       crime fiction stories too?.
   D. No, it was not made into a film. However, it was adapted
       into a popular Broadway musical.
   E. I don't know.
   F. None of the above.
```

Listing 5: Simple Synthetic Dataset example

```
Question: Which of the following options corresponds to "
    optimal "?
Options:
   A. ion.
   B. optimal.
   C. coins.
   D. jackie.
   E. I don't know.
   F. None of the above.
```

## A.3 PROMPT TEMPLATES AND EXAMPLES

Variable parts are highlighted in **bold**; whitespace placing is marked by underscores; the position of line breaks is explicitly shown by symbols '\n' (note that the last line always ends without whitespace or line break). In our datasets, we ensured that each question ends with a question mark, and each choice ends with a point (a single whitespace before it does not affect the logic of tokenization by the LLaMA tokenizer).

Listing 6: MMLU prompt template

```
Question:␣{Text of the question}?\n
Options:\n
A.␣{Text of the option A}␣.\n
B.␣{Text of the option B}␣.\n
C.␣{Text of the option C}␣.\n
D.␣{Text of the option D}␣.\n
E.␣I␣don't␣know␣.\n
F.␣None␣of␣the␣above␣.\n
Answer:
```

Listing 7: CosmosQA/HellaSwag/Halu Dialogue prompt template

```
Context:␣
   {The context of the question/situation or the dialog history}
   \n
Question:␣{Text of the question}?\n
Options:\n
```

```
A._{Text of the option A}_.\n
B._{Text of the option B}_.\n
C._{Text of the option C}_.\n
D._{Text of the option D}_.\n
E._I_don't_know_.\n
F._None_of_the_above_.\n
Answer:
```

The following is an example of a 1 shot prompt from MMLU. 2-3-4-5-shot prompts were built in the same way, and prompts for datasets with context were built the same way, except each question is preceded by its context. Note that in demonstrations, we add a single whitespace between "`Answer:`" and the correct choice letter; for example, "`Answer:    A`", but *never* "`Answer:A`". This is done because sequences like "`:    A`" and "`:A`" are differently split into tokens by the LLaMA tokenizer. The former produces the same tokens corresponding to the letter "`A`" as in the choice option line, while later yields a different version of "`A`". From LLaMA's point of view, these two versions of letters are separate entities and are NOT interchangeable. Removing those symbols of whitespace often leads to a noticeable drop in performance.

Listing 8: An example of 1-shot prompt for a question from MMLU dataset

```
Question: A medication prescribed by a psychiatrist for major
    depressive disorder would most likely influence the balance
     of which of the following neurotransmitters?\n
Options:\n
A. serotonin .\n
B. dopamine .\n
C. acetylcholine .\n
D. thorazine .\n
E. I don't know .\n
F. None of the above .\n
Answer: A\n
Question:
    Meat should be kept frozen at what temperature in Fahrenheit?
    \n
Options:\n
A. 0 degrees or below .\n
B. between 10 and 20 degrees .\n
C. between 20 and 30 degrees .\n
D. 0 degrees or below .\n
E. I don't know .\n
F. None of the above .\n
Answer:
```

## B  OPTION-REPRESENTATIVE TOKENS ANALYSIS

For our method, it is crucial to identify particular option-representative tokens $\{t_i\}$ that concentrate the semantic information of each option. Given the causal nature of the attention mechanism in LLMs, the most logical choice is the last token following the option's content—typically the "\n" token. We use this in most of our experiments, though other tokens are also worth analysing: the label itself, the period after the label, and the period after the option content (see Figure 2 left). We also tested the mean aggregated score across all tokens in the option's content, but this approach yielded poor results. A detailed analysis of these variations for attention scores is shown in the right part of Figure 2. Our findings indicate that the period after the content and the end-of-line token are the most representative for our scores and the task. Interestingly, while the label token is almost useless in the zero-shot setup, which is consistent with (Lieberum et al., 2023), it performs well in the five-shot setup for certain heads. We hypothesise there exist multiple types of select-and-copy heads, each influencing the logits in distinct ways.

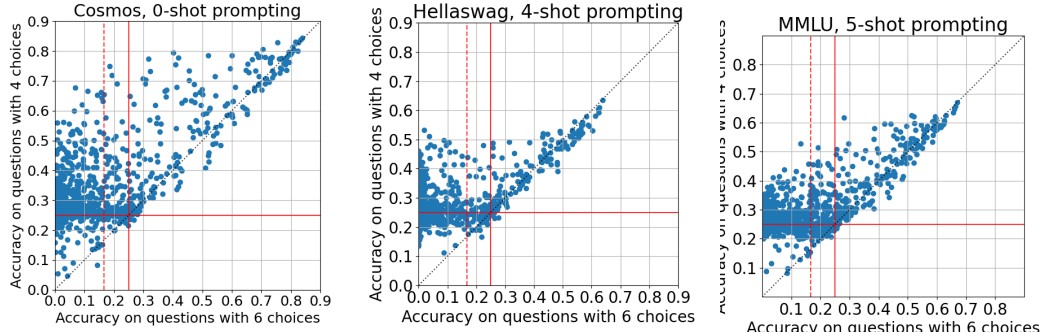

Figure 8: Correlation between heads QK-scoring accuracy on questions with 4 ('A'-'D') and 6 ('A'-'F') answer options. Solid red lines mark the accuracy level of $0.25$, dashed red line – $0.167$ (6 options random choice accuracy). Model LLaMA3.1-8B (base).

## C  SOME MORE INTUITION ON OPTIONS 'E' AND 'F'

As mentioned in the main text, including fictional, though always incorrect, choices "`E. None of the above`" and "`F. I don't know`" in every question was aimed at creating the "uncertainty sinks". However, they are also beneficial for analyzing attention head roles, but that is somewhat beyond the scope of this article. Here, we would like to provide some intuition about it.

We performed experiments on a modified version of our datasets, where questions include only 4 "meaningful" choices, i.e., options 'A'-'D' only. Scatterplots in Figure 8 show the correlation between the accuracy of heads using QK-scores on options without 'E'-'F' (by y-axis) and their accuracy on questions with all six options (by x-axis). Here, only validation subsets were used. We present plots for some possible setups, but others follow similar patterns. From these charts, we can see that if a head reaches good accuracy answering 4-choice questions, it usually will reach nearly the same accuracy on questions with six choices and vice versa; see points around the diagonal $y = x$ in the upper-right quadrant.

We can also observe another significant trend: horizontal stripe near y-level $0.25$. It can be explained in the following manner: in the data used, ground-truth answers are perfectly balanced – that is, for every choice 'A'-'D' $25\%$ of the questions have it as the correct answer. Therefore, if a head reaches 4-choice accuracy of $\approx 25\%$, it falls into one of the three categories:

1. This head chooses only one option in all questions. Usually, it is the last one on the list.
2. This head "guesses" answers, choosing options nearly randomly and "independent" from their meanings.
3. This head "understands" questions but is genuinely bad at answering them.

The addition of choices 'E' and 'F' drops the performance of the first type heads down to nearly $0\%$, second type – to around $16.7\%$; QK-scoring accuracy of the third type heads, however, usually remains the same.

Thus, we can conclude that choices 'E' and 'F' cause little effect on the performance of good heads, but, at the same time, their inclusion creates separation between heads that are bad at Multiple Choice Question Answering and heads that do not have MCQA in their functionality at all (they may perform other roles for LM).

Results on Figure 8 are given for LLaMA3.1-8B (base) model, but descibed patterns are typical for other models as well.

## D  PROMPT DESIGN ANALYSIS

Large Language Models are sensitive to prompt design and this is a well-known issue. Appendix A.3 provides the details on the prompt templates we used in our main experiments. Those prompts do not contain instructions, so the primary intent of this section is to investigate effect of added instructions.

First, we considered 3 various formal style instructions to make the task clearer for the model. For experiments on MMLU dataset we removed mentions of context from the instructions.

Listing 9: Explicit instructions

```
1. Make your best effort and select the correct answer for the
   following question based on the context. You only need to
   output the option.
2. Think logically and select the correct answer for the
   following question based on the context. You only need to
   output the option.
3.Answer the question based on the context. Select the option
   you are most confident at.
```

The questions prompted with instructions were built according to the modified template:

Listing 10: Modified template for prompt with instruction

```
Instruction\n
Context: \n [for MMLU this line is omitted]
Question: {Text of the question}?\n
...
```

We conducted experiments with 'instructioned' prompts in $0-$shot setup for several models. The setup was similar to the main experiments reported in this paper. The only difference was that for evaluation we used a 1,000-sample subset of the original evaluation set (it was done to reduce computational time; the class balance was preserved).

Normally, for each prompt individually, we selected the best head on the calibration data subset and measured the performance of QK-scores from that head on the evaluation data. However, we observed further stability of the QK-score method. Apparently, QK-score on heads chosen with the default prompts (i.e. without instructions), can achieve near the same level of performance even if prompt contains explicit instruction. And so, we took best heads found on calibration set with no-instruction prompts (e.g. head $(16, 19)$ for LLaMA3.1-8B on MMLU, and head $(17, 24)$ on Halu Dialogue, or head $(14, 24)$ for LLaMA2-7B on MMLU) and evaluated their quality on calibration sets when prompts are given with instructions. In the table this experiment is given in "QK-score + fixed head" rows. We report results only for LLaMA2-7B-base/-chat, but for other models the overall picture is similar.

The results are presented in Table 3. We see that in $0-$shot setup on all datasets clear instructions in the prompt improve the performance of both Baseline and QK-score methods compared to our default prompts. However, this increase is not uniform and is much more pronounced for chat-/instruct-tuned versions. Also, it can be noted that QK-score often has lower variance in both metrics. For the family of LLaMA2 models, QK-score with 'fixed head' achieves the same level of performance as QK-score with heads selected for each prompt individually. For the chat-tuned model there is some difference between them ('fixed head' slightly degrades the performance).

| Model | Method | | MMLU | Cosmos QA | Hellaswag QA | Halu Dialogue |
|---|---|---|---|---|---|---|
| **Qwen2.5 -1.5B** | Baseline | Acc | $59.5 \pm 0.6$ | $74.3 \pm 1.2$ | $58.1 \pm 2.2$ | $35.9 \pm 2.1$ |
| | | PA | $49.3 \pm 0.4$ | $67.8 \pm 1.6$ | $48.6 \pm 2.9$ | $25.7 \pm 1.7$ |
| | QK-score | Acc | $57.0 \pm 1.3$ | $72.1 \pm 0.9$ | $58.7 \pm 1.0$ | $38.3 \pm 0.9$ |
| | | PA | $45.9 \pm 1.1$ | $64.2 \pm 0.8$ | $48.6 \pm 0.3$ | $27.5 \pm 0.8$ |
| **LLaMA2 -7B** | Baseline | Acc | $29.2 \pm 1.5$ | $37.3 \pm 1.2$ | $29.4 \pm 0.2$ | $26.5 \pm 1.1$ |
| | | PA | $10.6 \pm 0.5$ | $17.3 \pm 0.9$ | $9.1 \pm 0.1$ | $7.2 \pm 1.2$ |
| | QK-score | Acc | $32.2 \pm 2.6$ | $51.9 \pm 0.3$ | $37.4 \pm 0.5$ | $35.6 \pm 0.3$ |
| | | PA | $13.5 \pm 1.9$ | $33.8 \pm 0.7$ | $18.4 \pm 1.6$ | $12.8 \pm 0.9$ |
| | + Fixed Head | Acc | $33.0 \pm 1.5$ | $51.9 \pm 0.3$ | $36.4 \pm 1.5$ | $34.4 \pm 1.5$ |
| | | PA | $15.9 \pm 1.0$ | $33.8 \pm 0.7$ | $17.9 \pm 2.0$ | $14.5 \pm 1.5$ |
| **LLaMA2 -7B, chat** | Baseline | Acc | $44.1 \pm 1.1$ | $62.1 \pm 0.1$ | $42.9 \pm 0.4$ | $25.8 \pm 1.6$ |
| | | PA | $28.6 \pm 0.8$ | $47.0 \pm 0.9$ | $25.0 \pm 1.5$ | $12.6 \pm 0.8$ |
| | QK-score | Acc | $44.9 \pm 0.4$ | $64.0 \pm 1.2$ | $49.5 \pm 0.6$ | $44.0 \pm 0.5$ |
| | | PA | $30.1 \pm 1.2$ | $51.0 \pm 1.4$ | $34.6 \pm 1.1$ | $25.2 \pm 0.8$ |
| | + Fixed Head | Acc | $44.5 \pm 0.8$ | $62.8 \pm 0.4$ | $44.4 \pm 0.8$ | $35.8 \pm 0.6$ |
| | | PA | $28.9 \pm 0.1$ | $46.3 \pm 0.9$ | $30.1 \pm 1.5$ | $16.6 \pm 0.8$ |
| **LLaMA3 -8B, Instruct** | Baseline | Acc | $61.1 \pm 0.5$ | $84.5 \pm 0.5$ | $67.9 \pm 0.8$ | $60.7 \pm 0.5$ |
| | | PA | $52.4 \pm 0.6$ | $79.3 \pm 0.7$ | $59.3 \pm 0.9$ | $51.1 \pm 1.0$ |
| | QK-score | Acc | $61.5 \pm 0.1$ | $88.3 \pm 0.1$ | $70.8 \pm 1.9$ | $65.3 \pm 1.4$ |
| | | PA | $52.9 \pm 0.0$ | $84.4 \pm 0.2$ | $61.8 \pm 2.5$ | $53.5 \pm 4.4$ |
| **LLaMA3.1 -8B** | Baseline | Acc | $60.5 \pm 0.4$ | $78.7 \pm 0.9$ | $51.7 \pm 1.6$ | $55.3 \pm 1.7$ |
| | | PA | $46.8 \pm 0.2$ | $70.9 \pm 0.8$ | $35.5 \pm 3.0$ | $41.2 \pm 1.7$ |
| | QK-score | Acc | $64.2 \pm 0.6$ | $82.6 \pm 0.5$ | $67.9 \pm 0.4$ | $58.5 \pm 2.9$ |
| | | PA | $53.8 \pm 0.4$ | $76.6 \pm 0.8$ | $57.4 \pm 1.1$ | $42.7 \pm 3.0$ |
| | + Fixed Head | Acc | $64.2 \pm 0.6$ | $82.6 \pm 0.5$ | $58.4 \pm 0.9$ | $54.9 \pm 1.8$ |
| | | PA | $53.0 \pm 0.7$ | $75.4 \pm 0.6$ | $45.0 \pm 1.3$ | $39.4 \pm 1.9$ |
| **LLaMA3.1 -8B, Instruct** | Baseline | Acc | $66.1 \pm 1.5$ | $88.0 \pm 1.1$ | $71.8 \pm 1.8$ | $57.1 \pm 5.2$ |
| | | PA | $54.1 \pm 1.5$ | $83.5 \pm 1.5$ | $62.3 \pm 2.8$ | $47.5 \pm 5.3$ |
| | QK-score | Acc | $67.0 \pm 0.6$ | $89.7 \pm 0.6$ | $72.0 \pm 1.0$ | $69.7 \pm 1.8$ |
| | | PA | $57.6 \pm 1.2$ | $85.5 \pm 1.1$ | $63.6 \pm 1.8$ | $55.3 \pm 2.2$ |
| | + Fixed Head | Acc | $67.1 \pm 0.6$ | $89.9 \pm 0.4$ | $73.0 \pm 0.3$ | $71.0 \pm 1.6$ |
| | | PA | $57.1 \pm 0.6$ | $85.7 \pm 0.8$ | $65.1 \pm 0.5$ | $53.7 \pm 1.4$ |

Table 3: Performance of Baseline and QK-score methods on prompts with explicit instructions. 0-shot setup.

Next, we explore more exotic examples. As it was noted in Mozikov et al. (2024), behaviour of LLM's may change when models are prompted with different emotional states. Inspired by this work, we collected a set of 11 creative instructions: 8 instructions asking the model to emulate an emotion and 3 instructions asking to answer the question while roleplaying a fictional character. The setup of experiments was the same as the one used for explicit instructions. The results are presented in Table 4. Here we can see that, in general, our creative instructions result in the decrease of performance of both methods, and once again, this effect is more noticeable for chat-/instruct-tuned models.

As for prompts with clear instructions, evaluation performance of QK-score on best head fixed on the 'no-instruction' prompt is close to performance of QK-score on best head selected specifically for the prompt. This observation suggests that heads chosen for QK-scoring in our method are stable across a wide range of possible prompts (thus justifying the omission of clear instructions in our main experiments is justified).

| Model | Method | | MMLU | Cosmos QA | Hellaswag QA | Halu Dialogue |
|---|---|---|---|---|---|---|
| **Qwen2.5 -1.5B** | Baseline | Acc | $56.9 \pm 2.9$ | $71.4 \pm 2.7$ | $56.7 \pm 3.7$ | $38.4 \pm 2.5$ |
| | | PA | $45.9 \pm 3.2$ | $64.2 \pm 3.2$ | $47.3 \pm 3.8$ | $27.2 \pm 2.5$ |
| | QK-score | Acc | $55.3 \pm 2.0$ | $70.6 \pm 1.3$ | $56.1 \pm 1.8$ | $38.2 \pm 2.5$ |
| | | PA | $43.3 \pm 1.9$ | $61.3 \pm 1.4$ | $45.8 \pm 2.9$ | $23.6 \pm 5.7$ |
| **LLaMA2 -7B** | Baseline | Acc | $29.2 \pm 1.8$ | $29.6 \pm 5.0$ | $27.3 \pm 1.3$ | $23.9 \pm 1.2$ |
| | | PA | $10.0 \pm 1.5$ | $11.5 \pm 4.2$ | $8.1 \pm 0.9$ | $5.7 \pm 0.8$ |
| | QK-score | Acc | $28.2 \pm 1.8$ | $47.0 \pm 6.0$ | $38.3 \pm 1.6$ | $34.4 \pm 1.4$ |
| | | PA | $10.5 \pm 2.0$ | $27.6 \pm 7.2$ | $20.2 \pm 2.0$ | $13.7 \pm 2.1$ |
| | + Fixed Head | Acc | $28.9 \pm 4.6$ | $47.0 \pm 5.9$ | $33.5 \pm 4.6$ | $31.7 \pm 2.5$ |
| | | PA | $13.3 \pm 2.9$ | $28.3 \pm 5.9$ | $15.5 \pm 3.5$ | $13.6 \pm 2.2$ |
| **LLaMA2 -7B, chat** | Baseline | Acc | $38.2 \pm 6.4$ | $55.4 \pm 4.6$ | $36.7 \pm 4.5$ | $22.8 \pm 4.9$ |
| | | PA | $23.3 \pm 5.4$ | $39.2 \pm 5.8$ | $19.5 \pm 3.6$ | $10.7 \pm 2.5$ |
| | QK-score | Acc | $39.0 \pm 3.3$ | $55.4 \pm 4.6$ | $36.7 \pm 4.5$ | $40.0 \pm 3.0$ |
| | | PA | $22.6 \pm 4.8$ | $44.2 \pm 5.9$ | $25.7 \pm 3.7$ | $20.3 \pm 4.0$ |
| | + Fixed Head | Acc | $34.7 \pm 7.9$ | $57.8 \pm 3.9$ | $37.7 \pm 1.5$ | $34.8 \pm 3.4$ |
| | | PA | $21.2 \pm 5.3$ | $37.5 \pm 6.9$ | $22.0 \pm 2.1$ | $16.4 \pm 1.9$ |
| **LLaMA3 -8B, Instruct** | Baseline | Acc | $56.2 \pm 5.0$ | $80.0 \pm 4.1$ | $55.8 \pm 14.3$ | $55.2 \pm 9.2$ |
| | | PA | $47.6 \pm 4.7$ | $74.4 \pm 4.6$ | $46.5 \pm 14.2$ | $46.0 \pm 9.3$ |
| | QK-score | Acc | $60.8 \pm 1.8$ | $86.3 \pm 1.2$ | $64.8 \pm 5.9$ | $63.4 \pm 5.7$ |
| | | PA | $52.8 \pm 2.8$ | $81.5 \pm 2.0$ | $55.7 \pm 6.8$ | $52.6 \pm 7.8$ |
| **LLaMA3.1 -8B** | Baseline | Acc | $55.5 \pm 1.7$ | $75.6 \pm 1.5$ | $33.2 \pm 9.5$ | $49.6 \pm 4.1$ |
| | | PA | $43.5 \pm 2.3$ | $67.2 \pm 1.9$ | $18.2 \pm 6.9$ | $35.3 \pm 4.2$ |
| | QK-score | Acc | $62.9 \pm 1.1$ | $80.6 \pm 1.0$ | $62.2 \pm 2.2$ | $57.4 \pm 2.4$ |
| | | PA | $52.1 \pm 1.7$ | $73.4 \pm 1.3$ | $49.9 \pm 2.8$ | $42.1 \pm 3.5$ |
| | + Fixed Head | Acc | $63.2 \pm 1.0$ | $80.6 \pm 1.1$ | $47.8 \pm 5.6$ | $50.2 \pm 2.8$ |
| | | PA | $52.0 \pm 1.2$ | $72.1 \pm 1.3$ | $34.9 \pm 5.3$ | $35.2 \pm 3.3$ |
| **LLaMA3.1 -8B, Instruct** | Baseline | Acc | $60.9 \pm 6.3$ | $82.7 \pm 4.1$ | $54.9 \pm 18.3$ | $42.2 \pm 14.2$ |
| | | PA | $49.8 \pm 5.9$ | $77.6 \pm 4.4$ | $44.4 \pm 17.7$ | $32.9 \pm 13.3$ |
| | QK-score | Acc | $65.5 \pm 2.0$ | $88.2 \pm 1.1$ | $64.1 \pm 8.1$ | $64.9 \pm 5.6$ |
| | | PA | $55.9 \pm 2.2$ | $83.3 \pm 1.1$ | $53.8 \pm 8.5$ | $48.6 \pm 6.2$ |
| | + Fixed Head | Acc | $65.5 \pm 2.0$ | $88.0 \pm 1.2$ | $63.6 \pm 10.4$ | $65.6 \pm 5.7$ |
| | | PA | $55.6 \pm 1.8$ | $82.8 \pm 1.7$ | $54.1 \pm 10.0$ | $44.8 \pm 7.5$ |

Table 4: Performance on prompts asking to emulate an emotion (or roleplay a character) and then answer the question. 0-shot setup

# E  NUMERICAL RESULTS FOR COMPARISON OF QK-SCORE WITH OTHER METHODS

Table 5 provides numerical results for our main experiments with QK-scores from heads of the LLaMA3.1-8B (base) model that are presented in Figure 3 in the main text. As can be seen from the table, for MMLU and CosmosQA PRIDE shows the best performance, while QK-score yields results slightly worse (though within reasonable tolerance). It can be explained by the fact that PRIDE was initially aimed at improving the model performance by increasing stability of its predictions, and QK-score is more an analysis tool than a method for evaluation improvement.

In the same way, Table 6 provides numerical results for the LLaMA2-7B (base) model for comparison.

| Method | | ...-shot prompting | | | | | |
|---|---|---|---|---|---|---|---|
| | | 0 | 1 | 2 | 3 | 4 | 5 |
| **MMLU** | | | | | | | |
| Baseline | Acc | 59.0 | 63.3 | **63.3** | 62.6 | 62.7 | 63.7 |
| | PA | 48.4 | 52.6 | 52.9 | 52.1 | 52.0 | 53.3 |
| PRIDE | Acc | 62.4 | **64.0** | 63.3 | 63.0 | **63.4** | **64.1** |
| | PA | 52.9 | **53.8** | 54.2 | 53.7 | 53.5 | **54.1** |
| Attention | Acc | 62.7 | 62.1 | 62.1 | 62.5 | 62.6 | 62.9 |
| score | PA | 51.0 | 51.1 | 50.9 | 52.0 | 50.4 | 52.3 |
| QK-score | Acc | **63.4** | 62.1 | 61.8 | 62.5 | 62.7 | 62.0 |
| | PA | 52.8 | 52.0 | 51.0 | 53.6 | 52.8 | 53.4 |
| Synthetic | Acc | 61.7 | 60.8 | 60.0 | 62.9 | 63.7 | 63.5 |
| | PA | **53.6** | 52.8 | 51.9 | 52.9 | **53.5** | 53.4 |
| **Cosmos QA** | | | | | | | |
| Baseline | Acc | 80.6 | 86.2 | 86.4 | 86.1 | 86.5 | 86.1 |
| | PA | 73.0 | 80.7 | 80.4 | 80.4 | 80.7 | 80.5 |
| PRIDE | Acc | **84.7** | **86.5** | **86.8** | **86.6** | **86.9** | **87.0** |
| | PA | **79.2** | **81.9** | **82.2** | **82.2** | **82.6** | **82.5** |
| Attention | Acc | 84.2 | 85.5 | 85.5 | 85.2 | 85.2 | 85.0 |
| score | PA | 76.6 | 80.1 | 80.0 | 77.3 | 78.0 | 77.9 |
| QK-score | Acc | 84.0 | **86.5** | 85.6 | 85.9 | 85.6 | 85.4 |
| | PA | 78.3 | 80.6 | 80.0 | 80.1 | 78.8 | 77.8 |
| Synthetic | Acc | 83.1 | 84.5 | 84.2 | 83.5 | 84.1 | 84.0 |
| | PA | 77.6 | 80.6 | 78.9 | 78.3 | 79.2 | 79.4 |
| **Hellaswag QA** | | | | | | | |
| Baseline | Acc | 38.8 | 60.7 | 56.5 | 65.1 | 63.2 | 63.6 |
| | PA | 22.2 | 46.2 | 41.2 | 51.6 | 49.8 | 50.8 |
| PRIDE | Acc | 63.0 | 62.5 | 61.1 | **67.7** | 64.8 | 63.7 |
| | PA | 50.9 | 49.4 | 47.6 | **56.7** | 53.0 | 52.1 |
| Attention | Acc | 59.5 | **64.1** | 64.7 | 66.8 | **66.1** | **64.9** |
| score | PA | 47.9 | **54.2** | **54.9** | 54.6 | 54.5 | 54.8 |
| QK-score | Acc | **63.4** | **64.1** | 63.4 | 64.2 | 65.7 | **64.9** |
| | PA | **53.6** | **54.2** | 53.8 | 55.6 | **56.5** | **55.8** |
| Synthetic | Acc | 49.1 | 51.0 | 60.0 | 62.5 | 63.7 | 63.5 |
| | PA | 37.7 | 38.1 | 52.2 | 55.0 | 55.0 | 54.5 |
| **Halu Dialogue** | | | | | | | |
| Baseline | Acc | 51.2 | 50.9 | 60.6 | 61.7 | 61.2 | 61.3 |
| | PA | 37.0 | 39.2 | 49.6 | 50.8 | 47.7 | 49.1 |
| PRIDE | Acc | **58.3** | 51.5 | 61.1 | 62.5 | 62.5 | 62.3 |
| | PA | **47.4** | 40.1 | 50.9 | 52.8 | 51.6 | 52.3 |
| Attention | Acc | 56.6 | **56.3** | 65.2 | 67.6 | **68.6** | 67.3 |
| score | PA | 44.8 | **43.2** | **53.9** | 56.4 | **57.9** | 52.5 |
| QK-score | Acc | 57.0 | 54.5 | **65.2** | **67.7** | 67.6 | 67.0 |
| | PA | 42.6 | 41.6 | 49.9 | **57.0** | 53.7 | **53.1** |
| Synthetic | Acc | 51.3 | 51.0 | 60.1 | 62.9 | 63.6 | 62.9 |
| | PA | 37.5 | 39.4 | 50.2 | 52.7 | 53.7 | 53.0 |

Table 5: Comparison of different methods for LLaMA3.1-8B (base) on various Q&A datasets. Reported metrics are Accuracy (Acc) and Permutation Accuracy (PA). The best results are highlighted in **bold**.

# F BEST HEADS

We utilized the minimum accuracy percentiles to determine stable heads that can be seen in Figures 9 and 10. The head is considered "stable" if (1) *QK-score* gives both high average accuracy across tasks and (2) accuracy on each task is better than at least 90% of heads. From these figures, we can see that LLaMA-3-8B has more "stable" heads than LLaMA-2-7B, and LLaMA-3.1-8B, in turn, surpasses LLaMA-3-8B in the number of "stable" heads.

| Method | | ...-shot prompting | | | | | |
| --- | --- | --- | --- | --- | --- | --- | --- |
| | | 0 | 1 | 2 | 3 | 4 | 5 |
| **MMLU** | | | | | | | |
| Baseline | Acc | 26.7 | 39.1 | **43.1** | **43.7** | **44.1** | **43.8** |
| | PA | 8.9 | 21.3 | **26.2** | 27.4 | **28.5** | **28.4** |
| PRIDE | Acc | 15.5 | 36.9 | 39.8 | 40.8 | 41.5 | 42.7 |
| | PA | 5.7 | 20.8 | 24.2 | 24.6 | 25.6 | 28.9 |
| Attention | Acc | **34.8** | 39.9 | 39.8 | 40.5 | 41.0 | 42.1 |
| score | PA | **17.2** | 19.4 | 21.9 | 23.4 | 24.1 | 24.3 |
| QK-score | Acc | 33.6 | **40.7** | 42.1 | 40.5 | 42.0 | 42.7 |
| | PA | **17.2** | **21.7** | 23.7 | 22.0 | 23.4 | 24.0 |
| **Cosmos QA** | | | | | | | |
| Baseline | Acc | 31.1 | 39.3 | 59.1 | 56.9 | 57.9 | 54.7 |
| | PA | 11.1 | 21.9 | 44.1 | 39.2 | 40.7 | 35.7 |
| PRIDE | Acc | 15.2 | 44.6 | 58.6 | 59.2 | 60.7 | 61.3 |
| | PA | 6.8 | 25.7 | 42.7 | 43.7 | 45.7 | 46.3 |
| Attention | Acc | 40.6 | 48.5 | 60.9 | **61.8** | **62.3** | **62.3** |
| score | PA | 23.8 | 28.3 | 46.8 | **47.7** | **48.4** | **48.2** |
| QK-score | Acc | **41.4** | **50.0** | **61.5** | 59.3 | 61.5 | 61.0 |
| | PA | **25.6** | **33.6** | **47.3** | 43.6 | 47.1 | 46.4 |
| **Hellaswag QA** | | | | | | | |
| Baseline | Acc | 26.5 | 28.8 | 30.6 | 33.3 | 36.1 | 34.6 |
| | PA | 7.5 | 9.4 | 11.8 | 13.9 | 17.3 | 15.0 |
| PRIDE | Acc | 17.8 | 32.7 | 35.6 | 38.6 | 39.6 | 41.4 |
| | PA | 4.9 | 12.9 | 16.0 | 20.5 | 21.8 | 23.2 |
| Attention | Acc | **34.8** | **40.0** | **41.7** | 42.6 | 43.2 | **43.5** |
| score | PA | **18.3** | **22.9** | **24.9** | 27.2 | 26.6 | **26.5** |
| QK-score | Acc | 33.0 | 37.1 | 40.4 | **42.7** | **45.7** | 42.3 |
| | PA | 15.9 | 14.3 | 23.0 | 22.2 | **28.5** | 24.6 |
| **Halu Dialogue** | | | | | | | |
| Baseline | Acc | 21.1 | 30.9 | 34.2 | 36.1 | 34.5 | 35.6 |
| | PA | 5.4 | 10.2 | 14.3 | 18.9 | 16.8 | 20.7 |
| PRIDE | Acc | 3.0 | 32.0 | 35.5 | 36.3 | 36.5 | 36.1 |
| | PA | 0.5 | 12.8 | 18.2 | 17.7 | 18.9 | 20.8 |
| Attention | Acc | 31.4 | **39.9** | 39.3 | 41.1 | 42.1 | 39.9 |
| score | PA | 10.9 | **19.4** | 17.5 | 19.0 | 22.3 | 21.3 |
| QK-score | Acc | **37.1** | 36.6 | **40.6** | **42.3** | **45.3** | **42.8** |
| | PA | **17.7** | 14.6 | **19.6** | **22.0** | **25.9** | **22.2** |

Table 6: Comparison of different methods for LLaMA2-7B (base) on various Q&A datasets. Reported metrics are Accuracy (Acc) and Permutation Accuracy (PA). The best results are highlighted in **bold**.

| Dataset | Best (Layer, Head) |
| --- | --- |
| MMLU | **(16, 19)**, **(17, 24)**, (16, 26), (17, 26) |
| HaluDialogue | (14, 5), (14, 21), (14, 2), **(17, 24)** |
| HellaSwag | (14, 5) **(17, 24)**, (17, 29), **(16, 19)** |
| CosmosQA | **(17, 24)**, **(16,19)**, (16, 26), (17,29) |

Table 7: Common top 1% heads across all $n$-shots based on accuracy for different datasets for LLaMA-3.1-8B

For LLaMA-2-7B (Figure 10 (a)), the heads from the 14th layer show the highest accuracy on almost all percentiles again. We also listed the top 1% pairs for all datasets based on accuracy in Table 7. There is a noticeable overlap between heads for various setups, and, once again, all of them are in middle layers of the model.

| Dataset | Best (Layer, Head) |
|---------|-------------------|
| MMLU | **(14, 24)**, (15, 4), (17, 0), **(14, 20)**, |
| HaluDialogue | (14, 29), **(14, 24)**, (14, 26) |
| HellaSwag | (15, 5), (15, 4), (18, 10), **(14, 20)** |
| CosmosQA | **(14, 24)**, (15, 5), (15, 4), (15, 23), **(14, 20)** |

Table 8: Common top 1% heads across all $n$-shots based on accuracy for different datasets for LLaMA-2-7B

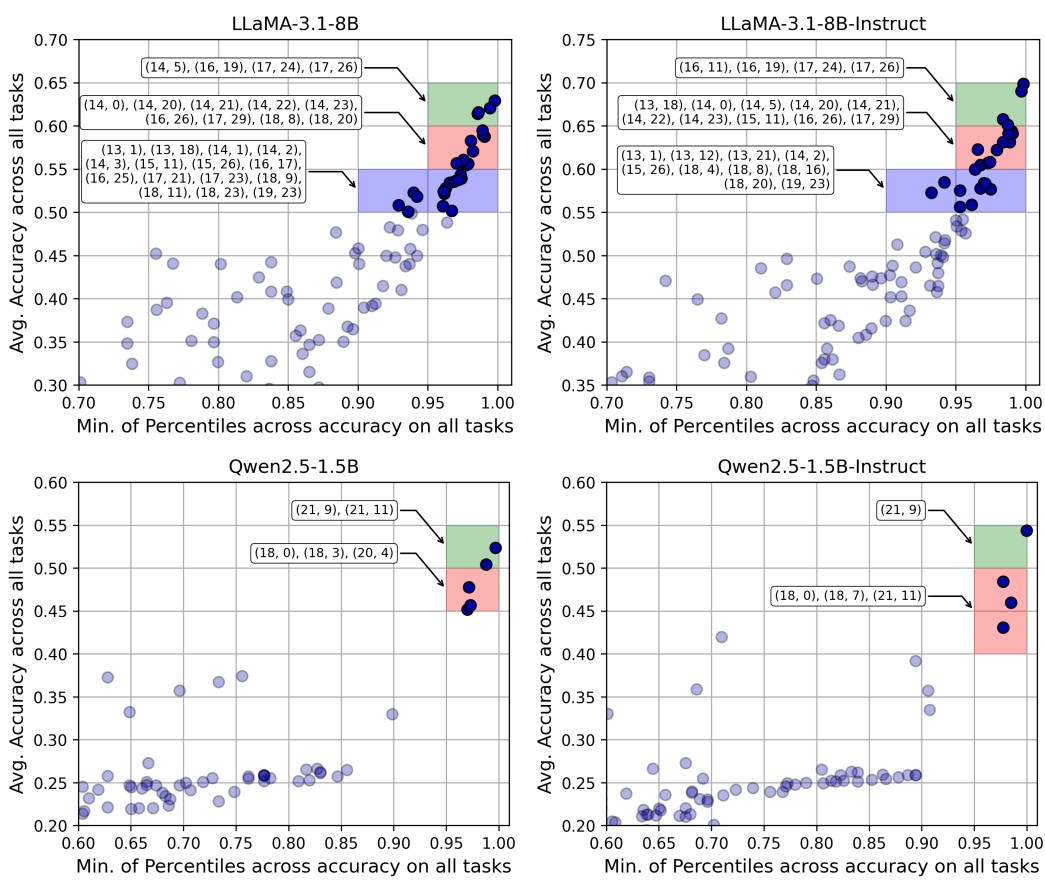

Figure 9: Stable heads for *QK-score* for 0-shot setup across all tasks for LLaMA-3.1-8B and Qwen2.5-1.5B (Base and Instruct). "$k$-th Minimum of Percentiles" means that the head is better than $k$ share of all heads for all tasks.

If we compare the performance of the "stable" heads with results obtained with preceding calibration in Figure 11, (14, 24) and (14, 20) are frequently chosen from the validation set. However, even when they do not, their performance is comparable to that of their validation-chosen counterparts, except for HaluDialogue. Besides, we tested the heads (14, 24), (14, 20), (14, 26), and (14, 13) for stability against increasing the number of options in SSD dataset (see Figure 14) and against changing the symbols that denote options, following Alzahrani et al. (2024) (see Appendix H). We also added other heads performing well on the SSD dataset to these plots for comparison.

For LLaMA-3.1-8B-base, most heads from the right-top corner of the left part of Figure 9 were already seen in the Figure 4 as heads in a green frames, i.e. top-5% across all four real datasets (namely, heads (16, 26), (18, 20), (14, 1), (14, 22), (17, 24), (16, 19), (14, 0), (18, 8), (14, 21), (17, 26), (19, 23), (17, 29), (18, 9), (14, 2), (14, 5), (14, 23), (14, 20)). Several of these heads also turned out to be very stable on the synthetic dataset, even when the number of options was increased (see Figure 14 for examples of such heads) and when the language was changed (see Table 10).

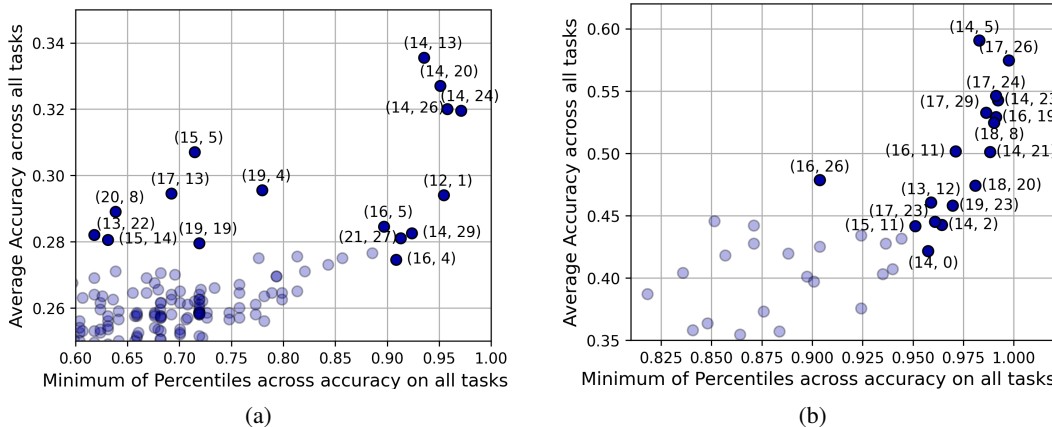

Figure 10: Stable heads for QK-score in (a) LLaMA2-7B and (b) LLaMA3-8B for 0-shot setup across all tasks. "$k$-th Minimum of Percentiles" means that the head is better than $k$ share of all heads for all tasks.

We also provided the results for LLaMA-3.1-8B-Instruct at the right part of the Figure 9 to compare the "stable head distribution" with the base version. Interestingly, there is a big intersection between best stable heads of base and Instruct versions of the model.

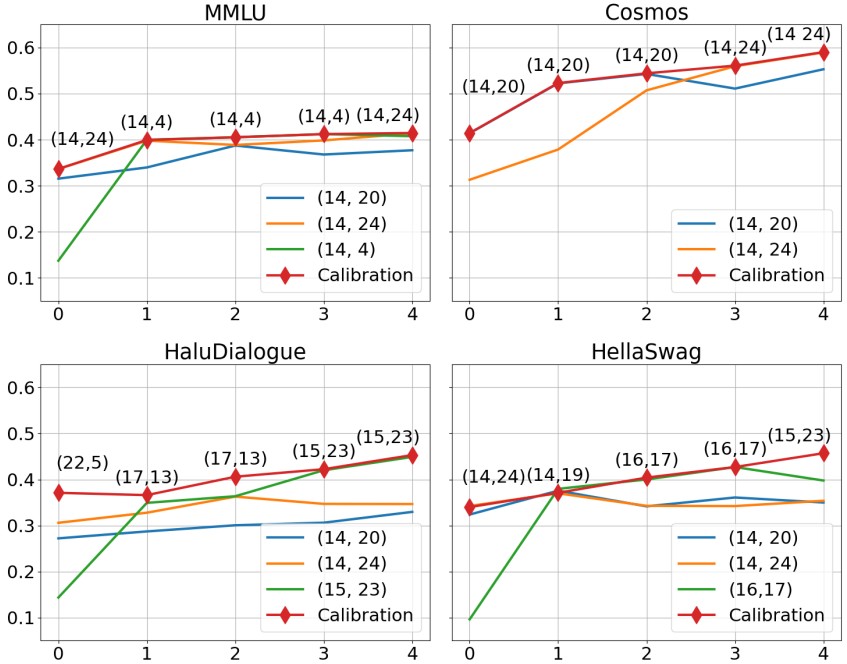

Figure 11: Accuracy of the best performing heads and several of the most robust heads – (14, 24), (14, 20) in LLaMA-2-7B

## G  HEADS ABLATION

In this section, we expand the paragraph **Select-and-copy heads ablation** of Section 7 by presenting the results of zero-ablation (Olsson et al., 2022) for the select-and-copy mechanism applied to two additional models: LLaMA-3.1-8B-Instruct (Fig. 12) and DeepSeek-R1-Distill-Qwen-7B (Fig. 13).

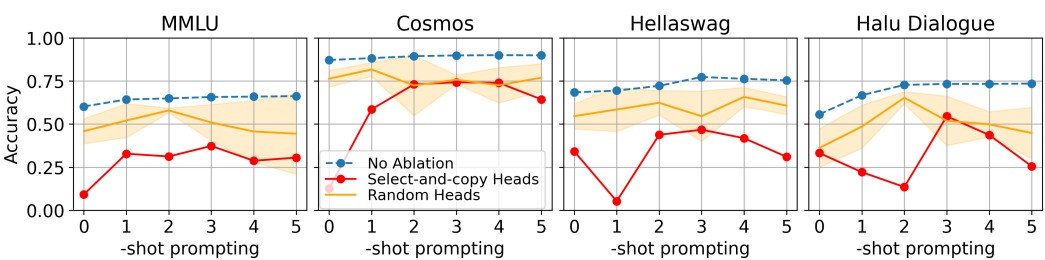

Figure 12: Zero-ablation of *select-and-copy* and random heads for LLaMA3.1-8B-Instruct

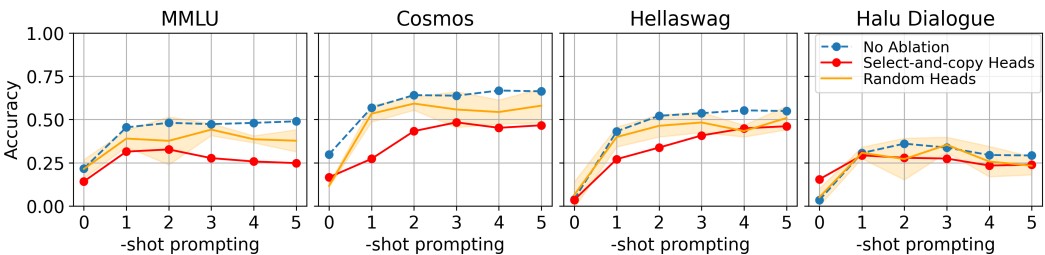

Figure 13: Zero-ablation of *select-and-copy* and random heads for DeepSeek-R1-Distill-Qwen-7B

We select attention heads based on above-random performance according to the QK-score, ablate them (zero out their outputs), and measure MCQA accuracy of the model (the *Baseline* setup) afterwards (red line on the plots, "Select-and-copy Heads"). For comparison, we also include results from random ablation, where an equal number of attention heads were ablated at random (yellow line, "Random Heads").

Random ablations were conducted across five independent runs, and we report the mean and standard deviation of model accuracy.

Although performance degradation due to zero-ablation remains substantial in finetuned and aligned models, in some cases it is comparable to that observed in random ablation. We hypothesize that this may suggest the involvement of other attention heads (associated with in-context learning or alignment) in the decision-making process.

## H BEHAVIOUR OF THE BEST HEADS UNDER THE CHANGE OF OPTIONS SYMBOLS AND OPTIONS AMOUNT

Aside from the standard version of the Simple Synthetic Dataset (SSD), which includes four essential options and two additional options, "E" and "F" (described in Section 5.1), we also considered alternative versions of the SSD with varying numbers of possible options. For instance, the version corresponding to the number "10" on the x-axis of Figure 5 contains ten essential options (A, B, C, D, E, F, G, H, I, J) and two special options: "K. I don't know" and "L. None of the above" (see Example 11). In these experiments, we used 200 examples from each version of the dataset to compute the attention scores.

Listing 11: Modification of SSD with ten options - example

```
Which of the following options corresponds to " mediterranean
    "?
Options:
    A: acceptance
    B: specialties
    C: charitable
    D: typically
    E: access
    F: jose
```

```
G: findlaw
H: colonial
I: mediterranean
J: data
K: I don't know.
L: None of the above.
```

Figure 14 is an extended version of Figure 5, showing more heads for LLaMA2-7B (left), LLaMA3-8B (center) and LLaMA3.1-8B (right). The heads for this figure are taken from the upper right sections of Figures 10 and 9, as they are the most stable across real datasets.

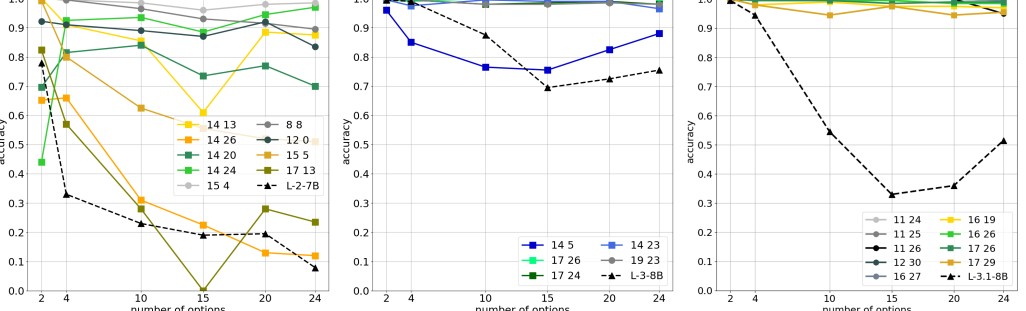

Figure 14: The results for various numbers of options in the Simple Synthetic Dataset (SSD) in a zero-shot setting are shown for LLaMA2-7B (left), LLaMA3-8B (center) and LLaMA3.1-8B (right). Different line colors represent the QK dot products from different heads. "Square" markers indicate heads that perform well across real datasets, while "round" markers represent heads that perform well on the synthetic dataset. Interestingly, more heads from newer versions of the LLaMA model show stability under increasing the number of options.

In Figures 15 and 16, we return to the standard 4-option SSD dataset but use different symbols for the option labels. The upper plot includes the renamed special options "E" for "I don't know" and "F" for "None of the above", while the lower plot omits them for the LLaMA2-7B model.

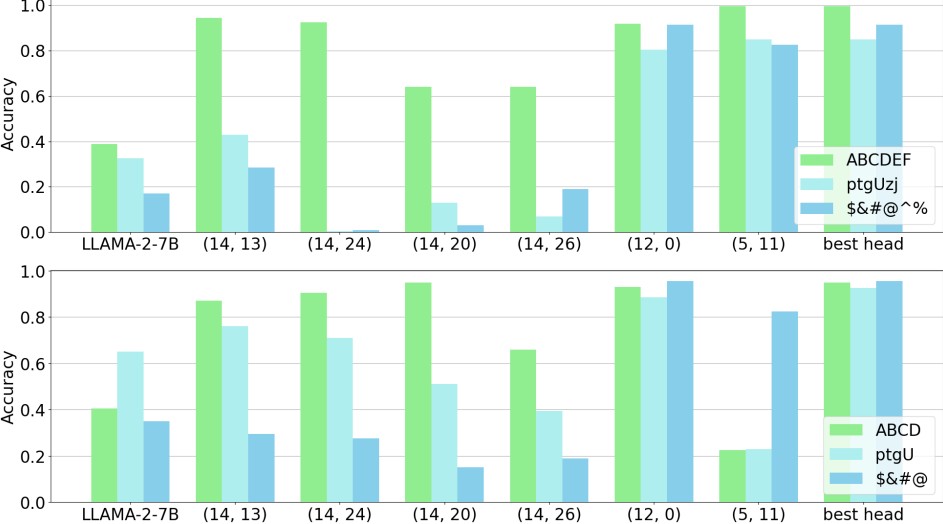

Figure 15: Performance of the QK-score from the best heads of the LLaMA2-7B for different option symbols, with "uncertainty" options (i.e. "I don't know" and "None of the above") presented (upper figure) and not presented (lower figure)

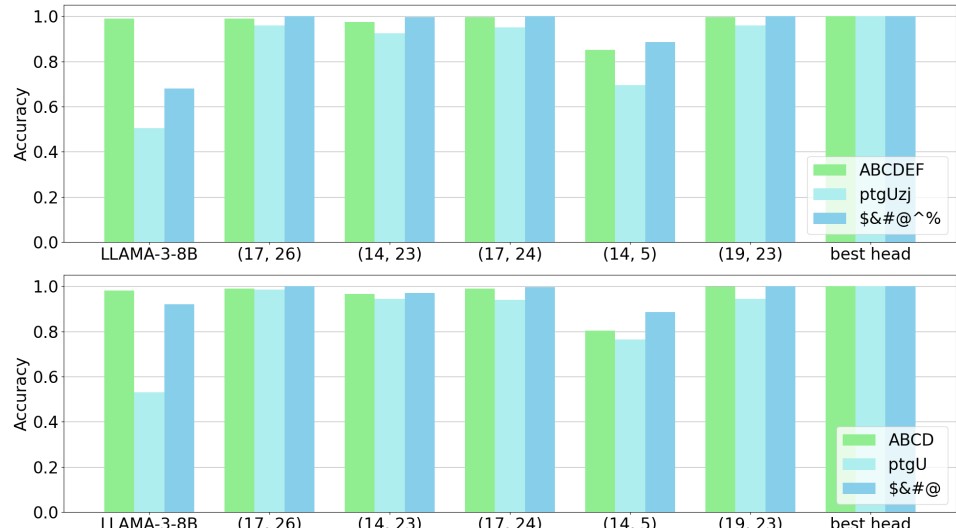

Figure 16: Upper row: performance of the QK-score from the best heads of the LLaMA3-8B for different option symbols, with "uncertainty" options (i.e. "I don't know" and "None of the above") presented (upper figure) and not presented (lower figure)

The accuracy of the best four heads of LLaMA-2-7B in Figure 10a declines in these new setups, but the head (12, 0) remains stable across all setups. Another interesting head is (5, 11): its accuracy is high for all setups with "uncertainty" and for "$&#" setup, but drops abruptly for "ABCD" and "ptgU". Studying such "anomalies" is a subject for future research.

We plot the results from the same setup for the LLaMA3-8B model in Figure 16. Interestingly, the best heads of the newer LLaMA3-8B model (see Figure 10b) exhibit significantly greater stability across the evaluated setups compared to those of the older LLaMA2 model.

# I   SYNTHETIC DATASET IN DIFFERENT LANGUAGES

We regenerated our synthetic dataset using three languages in addition to English. Figure 17 shows that the general distribution of QK-scores across heads of the LLaMA2-7B model on these multilingual datasets remains largely unchanged; for example, layers 8–15 still contain the most performant heads. However, differences in the performance of individual heads are also observed. Additionally, we show accuracies for top-10 performant heads in Table 9. We coloured green the heads that perform best across four real datasets (see Figure 10). Additionally, we highlighted in bold the heads that appear in the top-10 for all four languages.

As shown, 7 out of the top-10 best heads are shared across synthetic datasets in different languages, including two "green" heads that are also the best across our real datasets. This significant overlap suggests a substantial degree of universality among the identified heads. Interestingly, the QK-scores for the best heads are somewhat lower for English compared to the other languages we analysed. However, we cannot draw definitive conclusions from this observation without further investigation. A more thorough study of how QK-scores and the best-performing heads vary with the dataset's language remains a topic for future research and is beyond the scope of this paper.

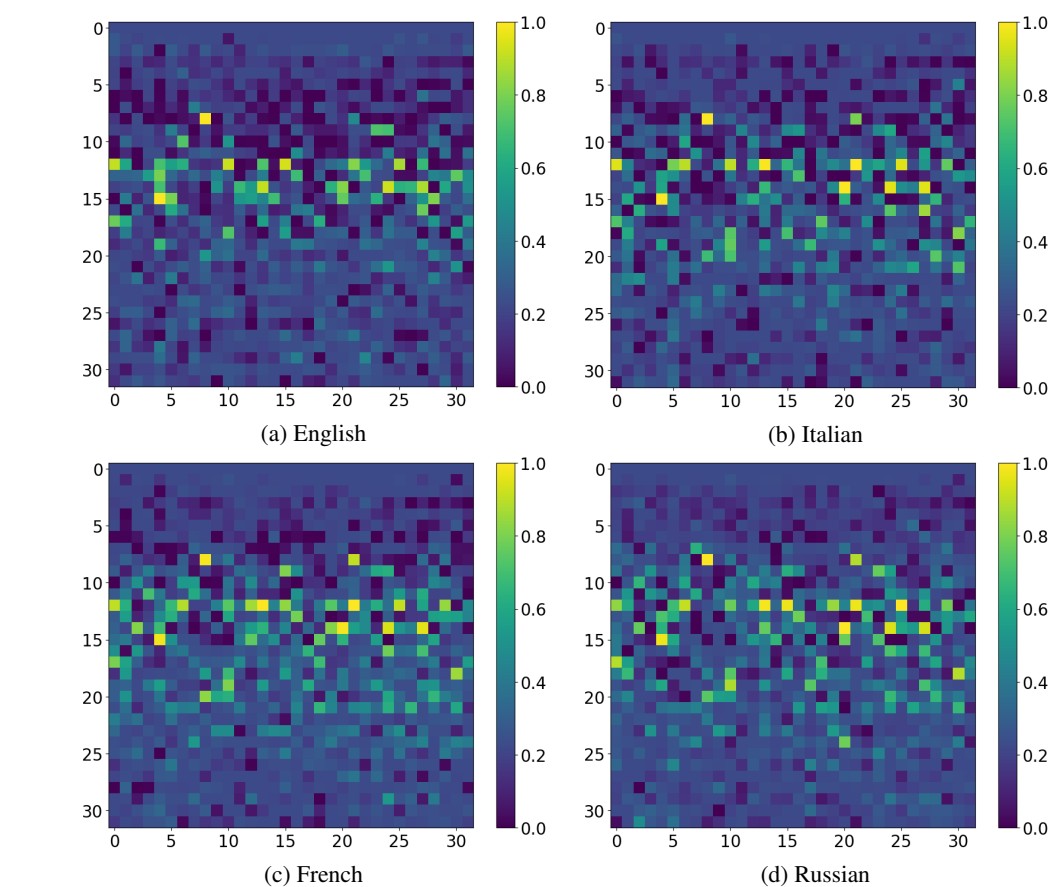

Figure 17: Performance of QK-score across different heads of LLaMA2-7B on a synthetic dataset generated in multiple languages

| Language | Best (Layer, Head) | Max Acc | Min Acc |
|---|---|---|---|
| English | **(8, 8)**, **(15, 4)**, (12, 15), **(14, 24)**, (12, 10) (14, 13), **(14, 27)**, **(12, 25)**, **(12, 21)**, **(14, 20)** | 0.995 | 0.815 |
| Italian | **(12, 21)**, **(15, 4)**, **(8, 8)**, **(14, 20)**, (12, 13) **(14, 24)**, **(14, 27)**, (12, 0), **(12, 25)**, (12, 10) | 1.000 | 0.900 |
| French | **(12, 21)**, (12, 13), **(8, 8)**, **(14, 20)**, **(15, 4)** **(14, 27)**, **(14, 24)**, (8, 21), **(12, 25)**, (12, 0) | 1.000 | 0.905 |
| Russian | **(12, 25)**, **(14, 20)**, **(8, 8)**, (12, 13), (12, 15) **(12, 21)**, **(15, 4)**, **(14, 24)**, **(14, 27)**, (12, 6) | 1.000 | 0.910 |

Table 9: Top-10 best heads per language for LLaMA 2-7B, sorted by decreased accuracy

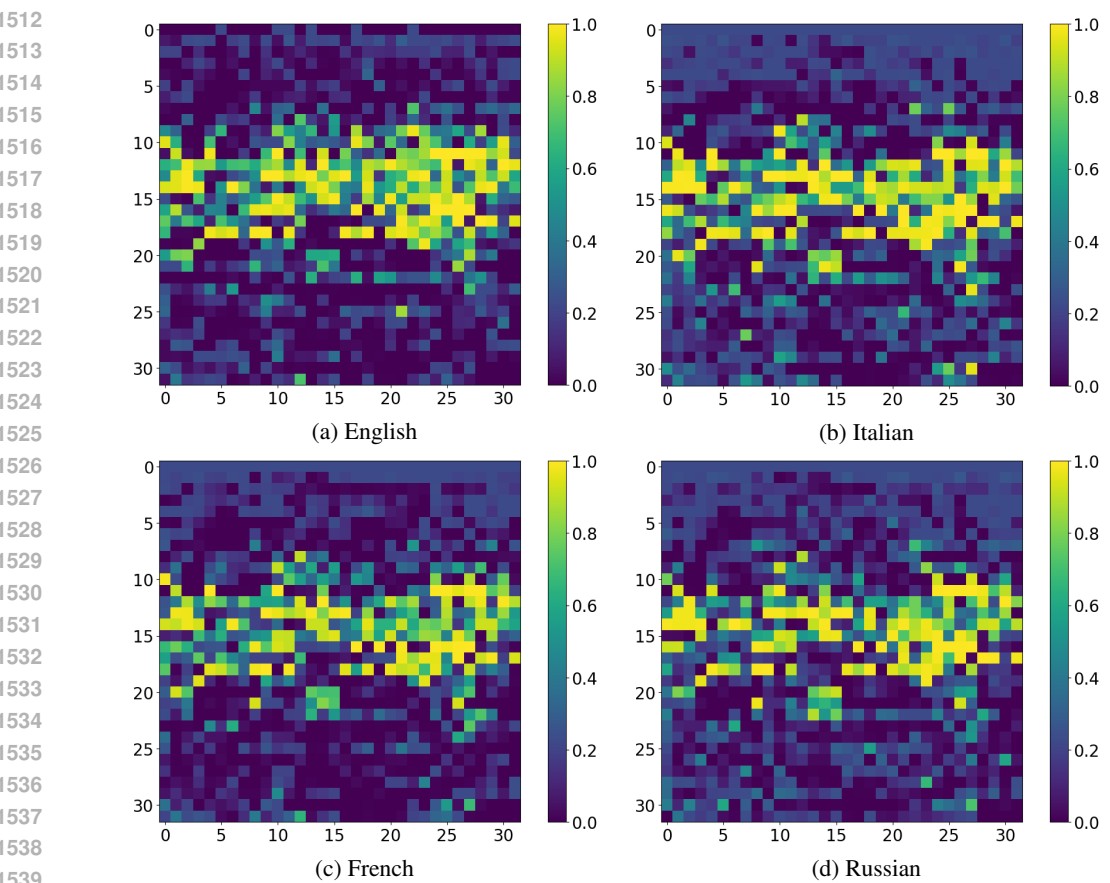

Figure 18: Performance of QK-score across heads of LLaMA3.1-8B on a multi-language SSD

| Language | Best (Layer, Head) | Max Acc | Min Acc |
|---|---|---|---|
| English | **(11, 24)**, **(11, 25)**, (11, 26), **(12, 30)**, (14, 6)
(15, 26), (16, 24), **(16, 27)**, (17, 28), (11, 10)
**(11, 27)**, (11, 28), (12, 14), (13, 2), **(13, 9)**
(13, 10), **(13, 16)**, (13, 23), (16, 17), (16, 25)
(16, 26), (17, 26), (18, 9), **(13, 13)**, (14, 2)
(15, 10), (17, 21), (18, 8), (18, 18), (12, 20) | 1.000 | 0.985 |
| Italian | **(11, 24)**, **(11, 25)**, **(11, 27)**, (12, 14), **(12, 30)**
**(13, 9)**, (13, 10), **(13, 13)**, (13, 15), **(13, 16)**
(14, 18), (15, 26), (16, 17), (16, 19), **(16, 27)**
(17, 21), (18, 9), (18, 18), (11, 4), (11, 10)
(11, 28), (13, 1), (14, 2), (14, 13), (14, 30)
(16, 10), (16, 11), (16, 21), (16, 26), (18, 24) | 1.000 | 0.995 |
| French | (11, 4), **(16, 27)**, (18, 9), **(11, 24)**, **(11, 25)**
**(11, 27)**, (11, 28), (12, 14), **(13, 9)**, (16, 17)
(16, 19), (16, 24), (16, 26), (17, 26), (11, 10)
**(12, 30)**, **(13, 16)**, (10, 0), **(13, 13)**, (14, 6)
(16, 21), (17, 29), (18, 11), (13, 1), (16, 8)
(16, 25), (17, 21), (18, 8), (12, 28), (13, 15) | 1.000 | 0.975 |
| Russian | **(11, 25)**, **(11, 27)**, (14, 6), **(16, 27)**, (17, 28)
**(11, 24)**, (11, 26), **(12, 30)**, **(13, 13)**, **(13, 16)**
(16, 19), (16, 24), (16, 25), **(16, 26)**, (17, 29)
**(18, 9)**, (13, 15), (14, 14), (14, 21), (16, 8)
**(16, 17)**, (17, 26), (18, 8), (18, 18), **(13, 9)**
(13, 10), (14, 0), (14, 3), (15, 20), (15, 24) | 1.000 | 0.985 |

Table 10: Top-30 best heads per language for LLaMA 3.1-8B, sorted by decreased accuracy

Figure 18 shows that similar patterns are observed for LLaMA3.1-8B, with the key difference being that the average quality of the heads in the middle layers is significantly higher than in LLaMA2-7B. Table 10 lists the top 30 heads for LLaMA3.1-8B, and again, we observe several heads that are universal across languages (marked in bold). As with LLaMA2-7B, we highlight in green the heads that perform best across four real-world datasets. We report the top 30 heads instead of the top 10 because LLaMA3.1-8B contains a large number of "good" heads with near-perfect performance – substantially more than LLaMA2-7B – making a top-10 selection unrepresentative for this model.

## J  BEST HEADS ON SYNTHETIC DATASET FOR QWEN 2.5-1.5B

In Figure 19, we present the accuracy of the QK-score for each head of Qwen 2.5-1.5B-base and -instruct on our synthetic datasets in four languages. These diagrams confirm that the layers with the best heads in Qwen 2.5-1.5B are closer to the final layer than in LLaMA-family models. Specifically, the best-performing heads in Qwen 2.5 are concentrated in layers 16–22, out of a total of 28 layers. Besides, earlier layers contain many heads with performance close to zero. Despite these differences, the overall pattern resembles the corresponding heatmaps for LLaMA-2-7B and LLaMA-3.1-8B shown in Figures 17 and 18, particularly in very early and late layers as both do not contain strongly pronounced select-and-copy heads. We also see that some especially "stable" heads, such as **(18, 3)**, **(20, 4)** and **(21, 11)**, which perform well across real datasets in Qwen 2.5-1.5B-base, remain among the best heads for the synthetic dataset in all languages for both Qwen 2.5-1.5B-base and Qwen 2.5-1.5B-Instruct models, as shown in Figure 19. This again illustrate the persistence of some select-and-copy heads across many datasets and languages, as discussed in our paper.

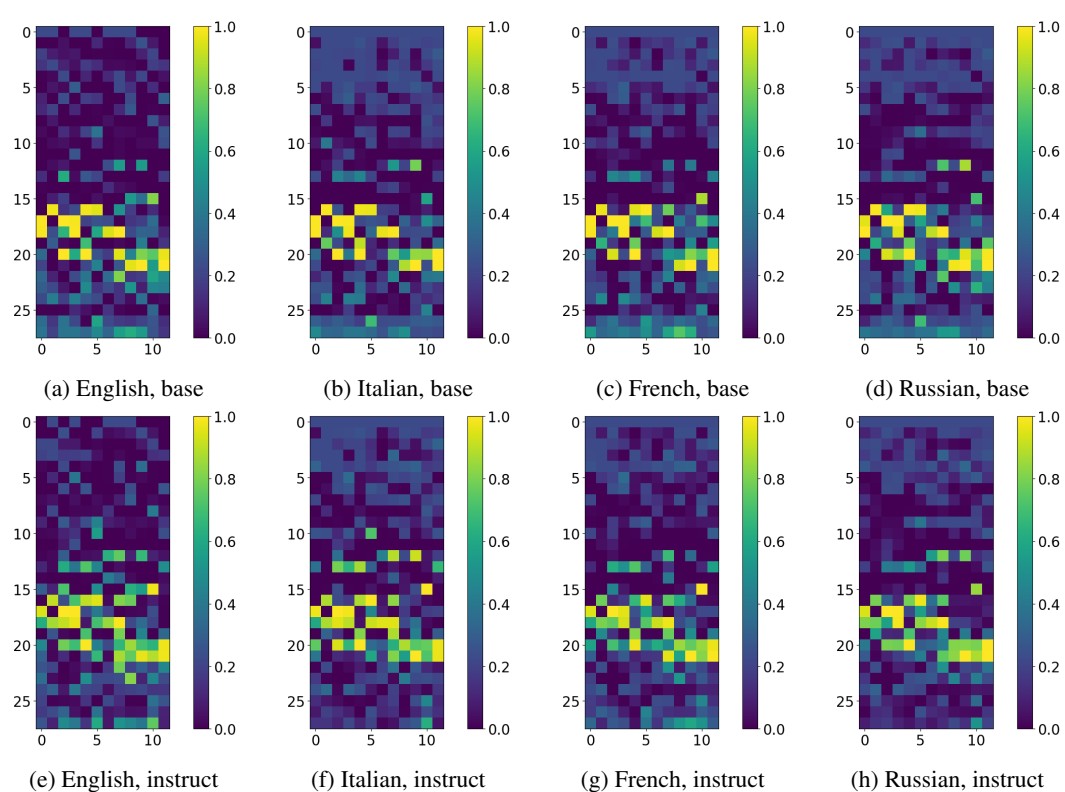

(a) English, base     (b) Italian, base     (c) French, base     (d) Russian, base

(e) English, instruct     (f) Italian, instruct     (g) French, instruct     (h) Russian, instruct

Figure 19: Performance of QK-score across different heads of Qwen 2.5-1.5B-base (upper row) and Qwen 2.5-1.5B-Instruct (lower row) on a multi-language SSD

# K HEAD SCORING WITHOUT VALIDATION SET

Let $\hat{\mathcal{D}}$ be some unlabelled MCQA dataset. Then, for each head we may calculate a score

$$HeadScore = \left(\frac{1}{|\hat{\mathcal{D}}|}\sum_{\hat{\mathcal{D}}}\sum_{i=1}^{n}a_{Nt_i}\right)\left(\frac{1}{|\hat{\mathcal{D}}|}\mathbb{I}\{\arg\max_i(a_{Nt_i}) \neq \hat{i}\}\right),$$

where $\hat{i}$ denotes the most frequent option for the given head; head indices $(l, h)$ are omitted. The left component represents the average amount of attention concentrated on the option-representative tokens $t_i, i = 1, \ldots, n$. The right component reflects the frequency of the situation, when the largest attention among the options falls on the option other than $\hat{i}$, i.e., any option other than the most frequent one.

The results of ranking heads according to these scores for LLaMA models are presented in Figure 20. We applied similar technique for Qwen-2.5-1.5B as well; the results are presented in the Figure 21. Here, we calculated the attention score without applying RoPE.

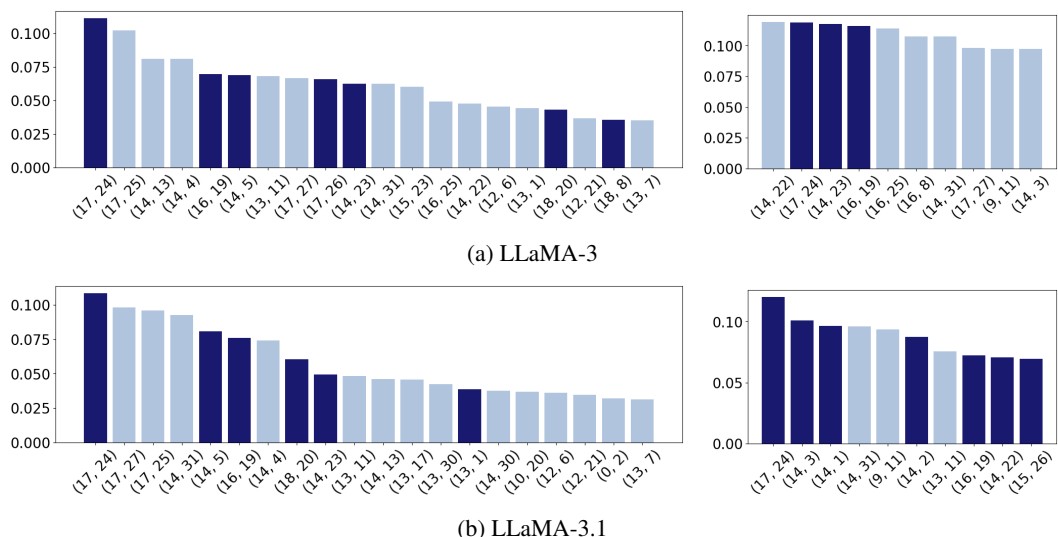

(a) LLaMA-3

(b) LLaMA-3.1

Figure 20: Left: average top heads scores of LLaMA models across real datasets (first twenty). Right: Top head scores on the Simple Synthetic Dataset (first ten). Dark blue marks the best heads, stable across all real datasets (i.e. heads from top-right corner of the Figures 9 and 10). Note that for calculating this score, we did not use the dataset labels.

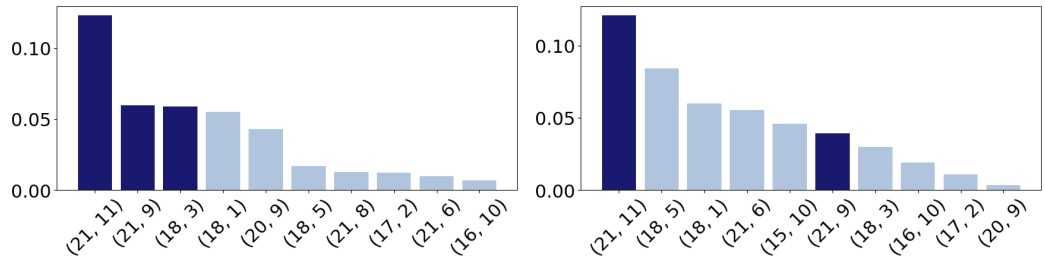

Figure 21: Left: average top heads scores for Qwen-2.5-1.5B-base across real datasets (first ten). Right: the same for Qwen-2.5-1.5B-instruct. Dark blue marks those heads that are the best across all four real datasets (see Figure 9).

For all three models (LLaMA-3-8B, LLaMA-3.1-8B and Qwen-2.5-1.5B), shown in the Figure 20, we see that the head with the highest unsupervised score, calculated on unlabelled examples from real

datasets, also exhibits consistently high accuracy across all these datasets, appearing in the top-right corner of Figures 9 or 10. Moreover, many other top-performing heads of these models also rank among those with the highest unsupervised scores on real and/or synthetic datasets.

## L  SELECTION BIAS

We investigate our methods in relation to the tendency to favor specific answer choices over the correct ones. Specifically, we compute how frequently the model itself or the *QK-score* across the top three attention heads selects each option. Ideally, an unbiased method should yield a near uniform distribution of predictions (i.e., its probability of correctly answering a question must be independent of the ground-truth option). Figure 22 illustrates the selection bias (expressed as a percentage) for the MMLU task in both 0-shot and 5-shot settings.

Our observations indicate that the models exhibit a distinct selection bias, and the *QK-score* sometimes reflects a similar distribution. However, the top heads often display differing distributions over the answer choices. Notably, in the five-shot setting, these distributions tend to align more closely with the distribution of correct answers.

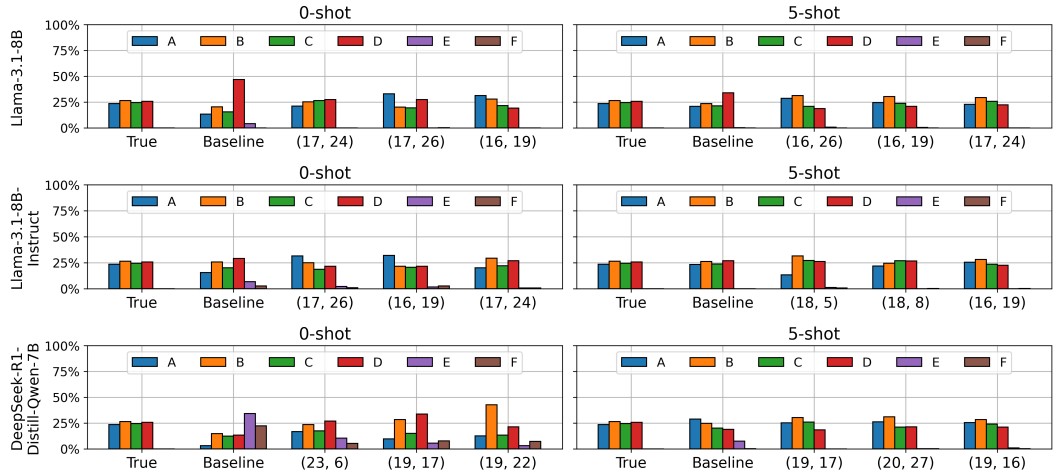

Figure 22: Distribution of predictions across options for different methods on MMLU 0-shot and 5-shot setup. $(l, h)$ depicts the distribution for $S_{QK}^{(l,h)}$

## M  COMPREHENSIVE RESULTS FOR EXPERIMENTS ON LARGER MODELS FROM LLAMA FAMILY

Here, we provide complete results of our experiments with QK-scores on four primary datasets (MMLU, CosmosQA, HellaSwag, and Halu Dialogue) for larger models. As before, the reported metrics are Accuracy and Permutation Accuracy.

- Figure 23 contains results for LLaMA2-13B, and Figure 30 for its chat-tuned version
- Figure 24 contains results for LLaMA2-70B, and Figure 31 for its chat-tuned version
- Figure 28 contains results for LLaMA-30B, and Figure 29 contains results for LLaMA-65B.
- Figure 25 contains results for LLaMA3-8B, and Figure 33 for its instruct-tuned version
- Figure 26 contains results for LLaMA3-70B, and Figure 34 for its instruct-tuned version
- Figure 27 contains results for LLaMA3.1-70B, and Figure 35 for its instruct-tuned version
- Figure 36 contains results for LLaMA3.3-70B-instruct

Our baseline accuracies are slightly lower than those in the original model reports (Touvron et al., 2023; Dubey et al., 2024) for two methodological reasons: (i) we evaluate with six answer options

(A–F) rather than the four used previously; and (ii) we benchmark the strict no-chain-of-thought setting, a regime that received limited coverage in the original papers.

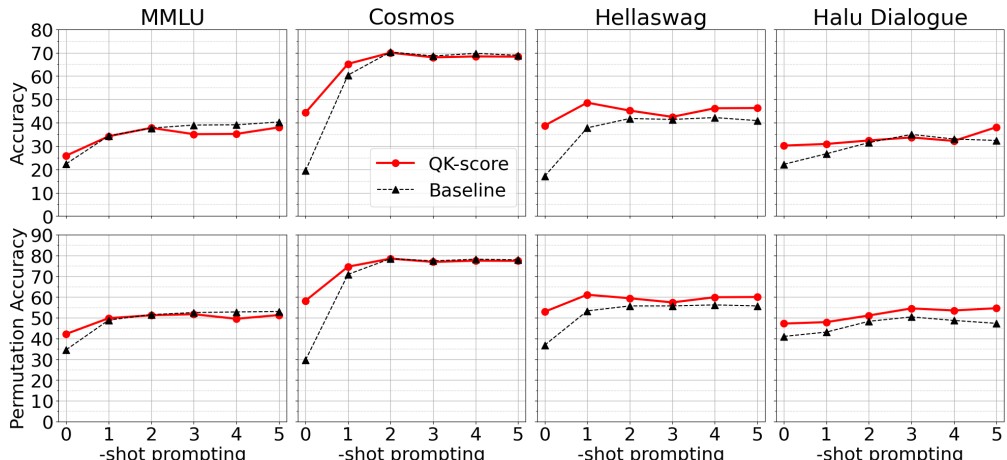

Figure 23: Comparison of different methods for LLaMA2-13B (base) on various Q&A datasets.

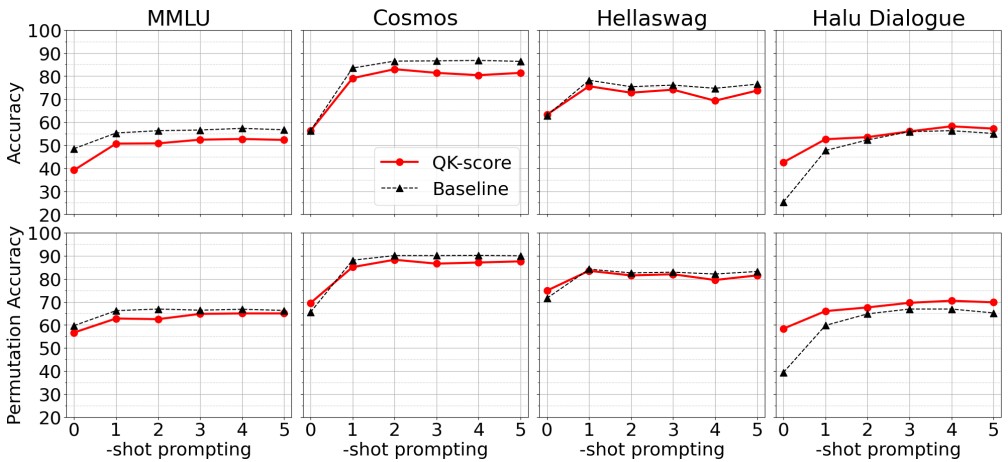

Figure 24: Comparison of different methods for LLaMA2-70B (base) on various Q&A datasets.

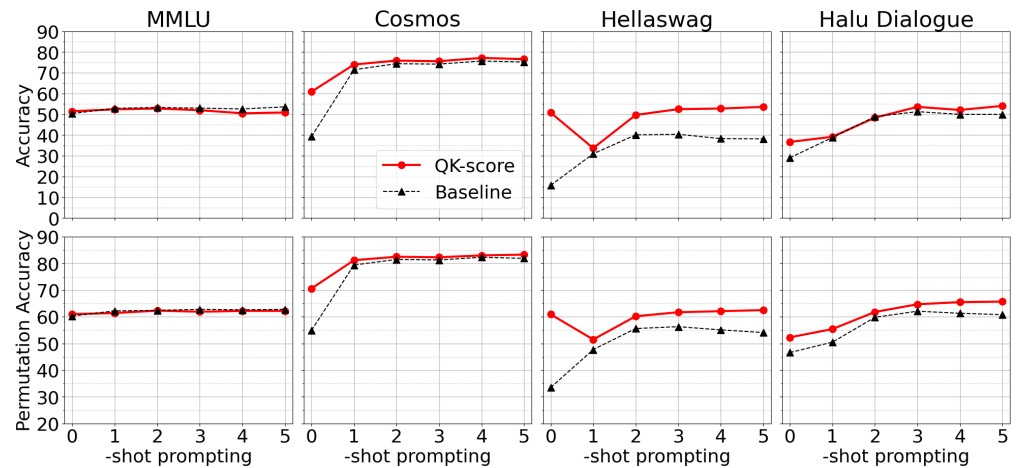

Figure 25: Comparison of different methods for LLaMA3-8B (base) on various Q&A datasets.

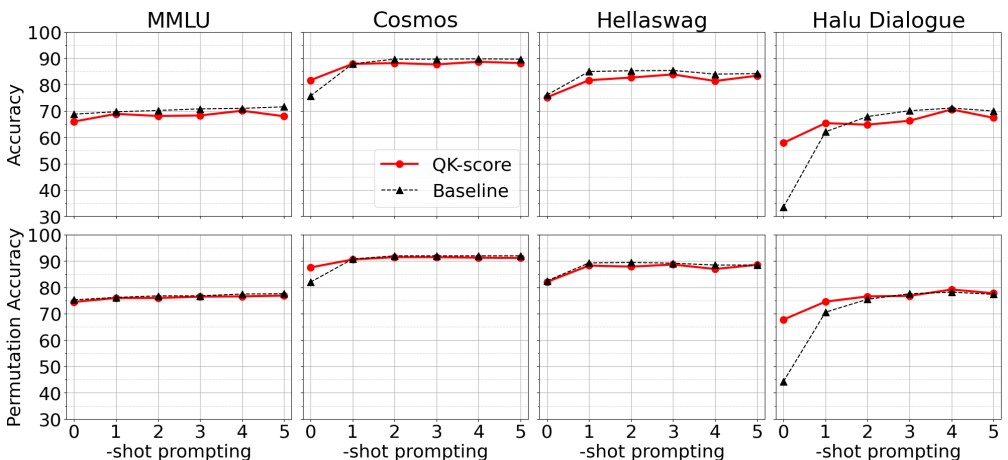

Figure 26: Comparison of different methods for LLaMA3-70B (base) on various Q&A datasets.

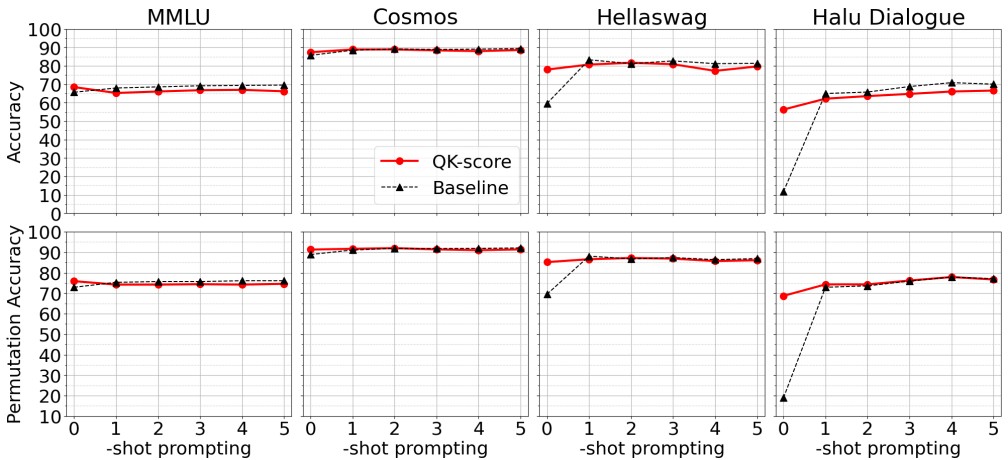

Figure 27: Comparison of different methods for LLaMA3.1-70B (base) on various Q&A datasets.

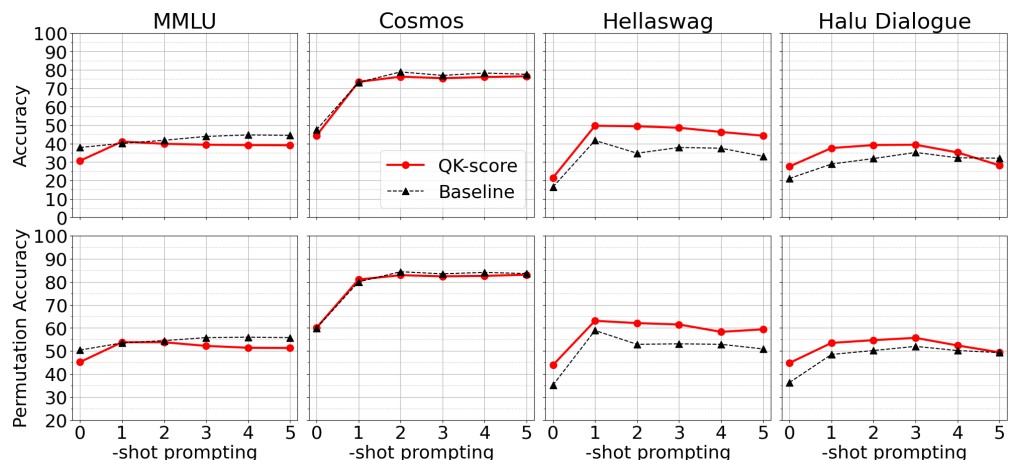

Figure 28: Comparison of different methods for LLaMA-30B (base) on various Q&A datasets.

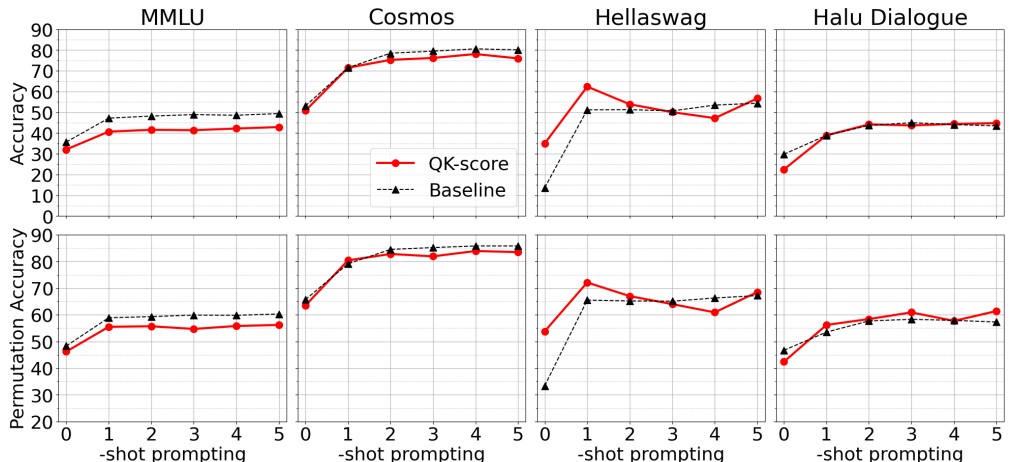

Figure 29: Comparison of different methods for LLaMA-65B (base) on various Q&A datasets.

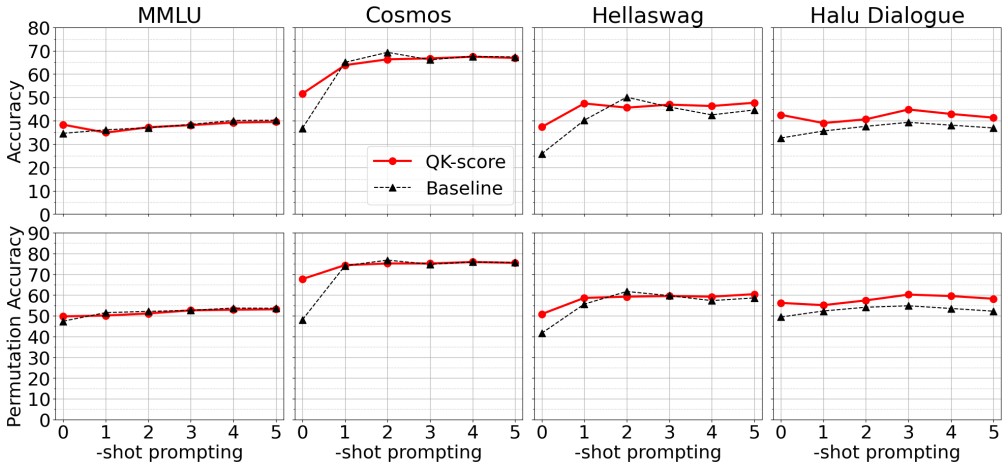

Figure 30: Comparison of different methods for LLaMA2-13B-chat on various Q&A datasets.

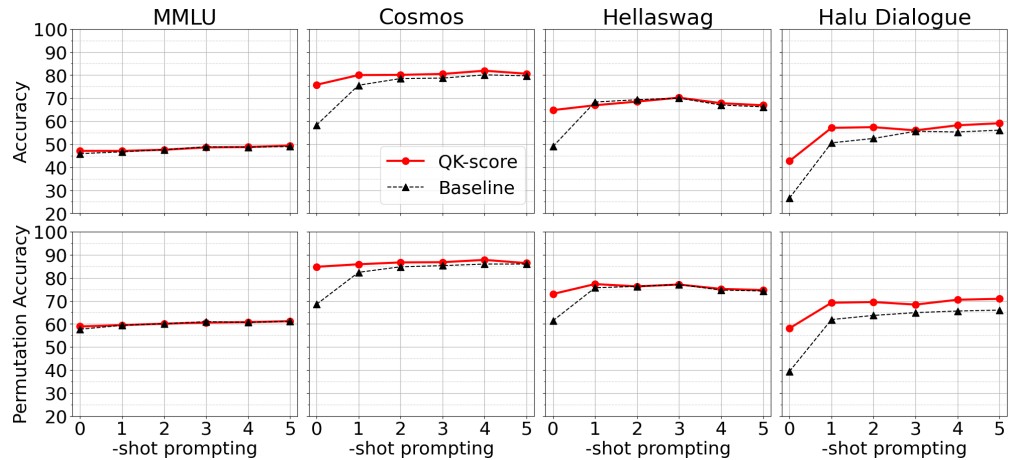

Figure 31: Comparison of different methods for LLaMA2-70B-chat on various Q&A datasets.

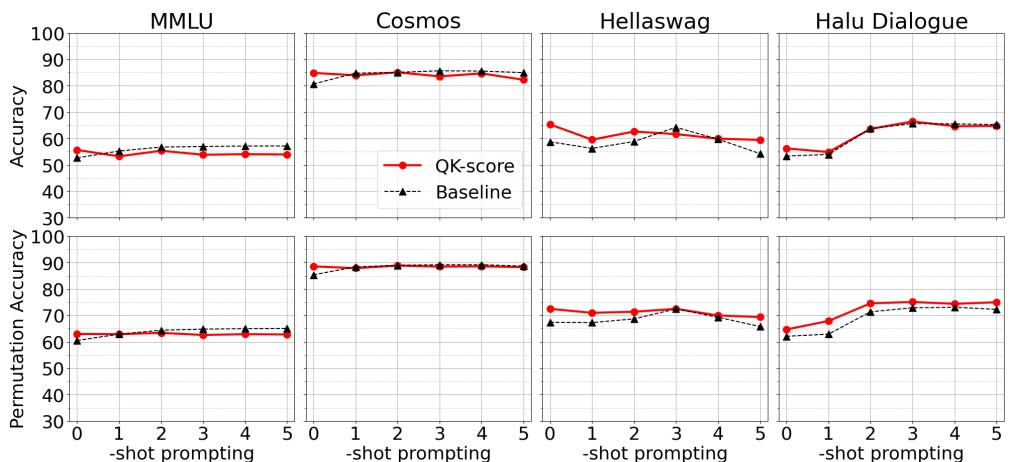

Figure 32: Comparison of different methods for LLaMA3-8B-instruct on various Q&A datasets.

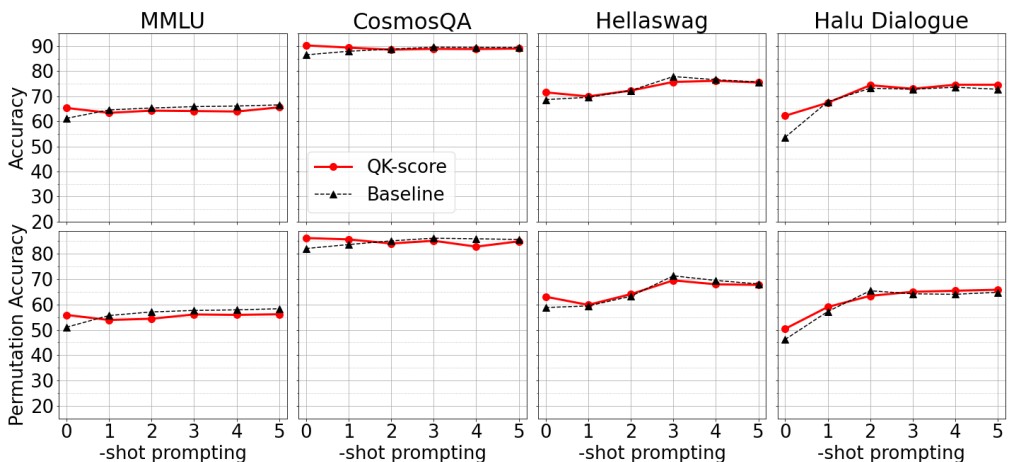

Figure 33: Comparison of different methods for LLaMA3.1-8B-instruct on various Q&A datasets.

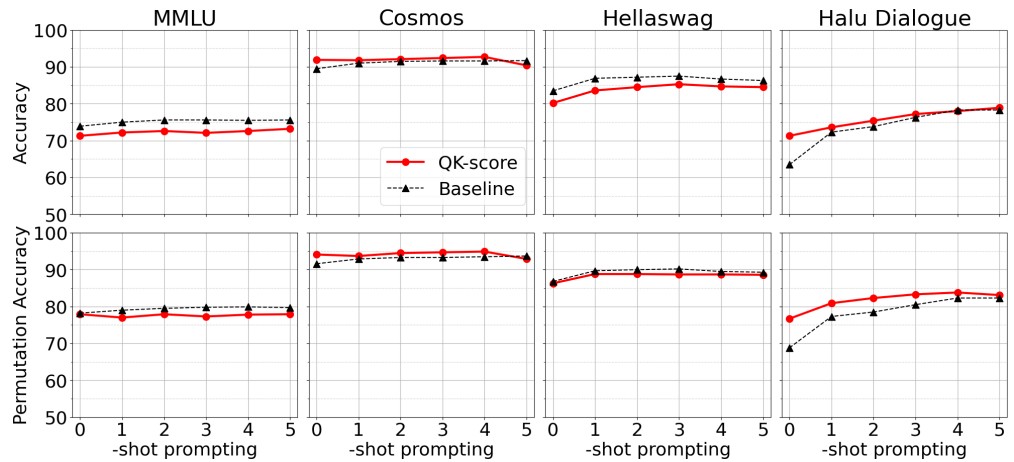

Figure 34: Comparison of different methods for LLaMA3-70B-instruct on various Q&A datasets.

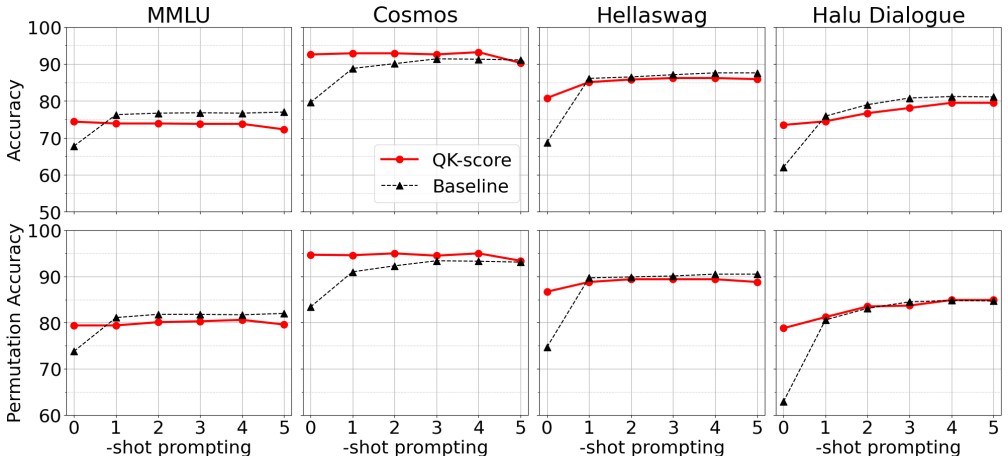

Figure 35: Comparison of different methods for LLaMA3.1-70B-instruct on various Q&A datasets.

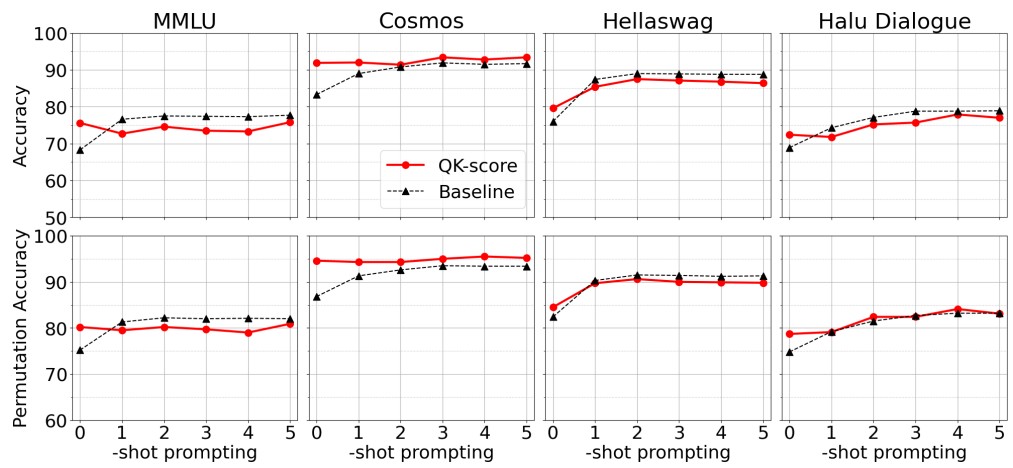

Figure 36: Comparison of different methods for LLaMA3.3-70B-instruct on various Q&A datasets.

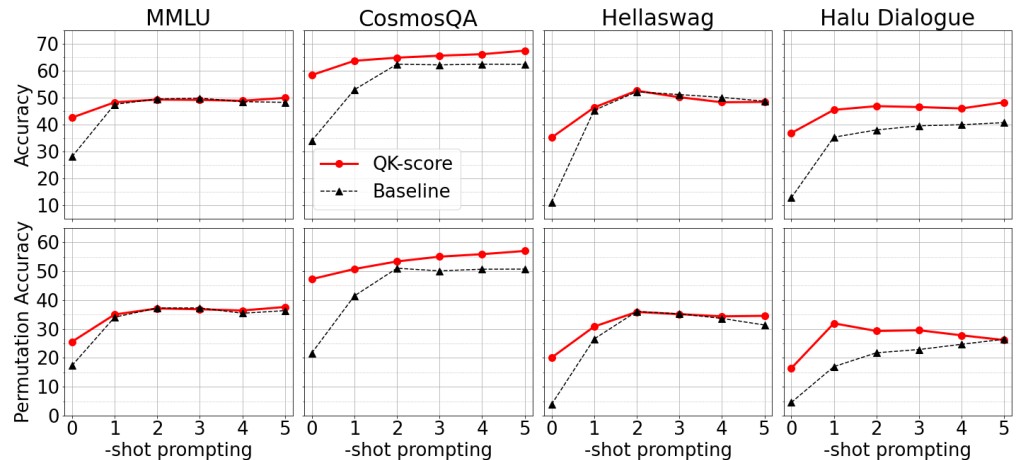

Figure 37: Comparison of different methods for DeepSeek-R1 distilled on Qwen2.5-7B on various Q&A datasets.

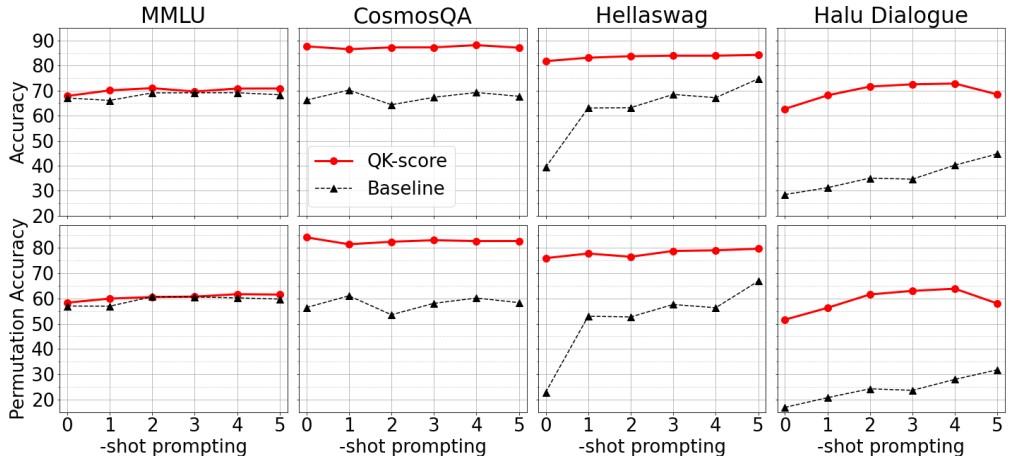

Figure 38: Comparison of different methods for DeepSeek-R1 distilled on Qwen2.5-14B on various Q&A datasets.

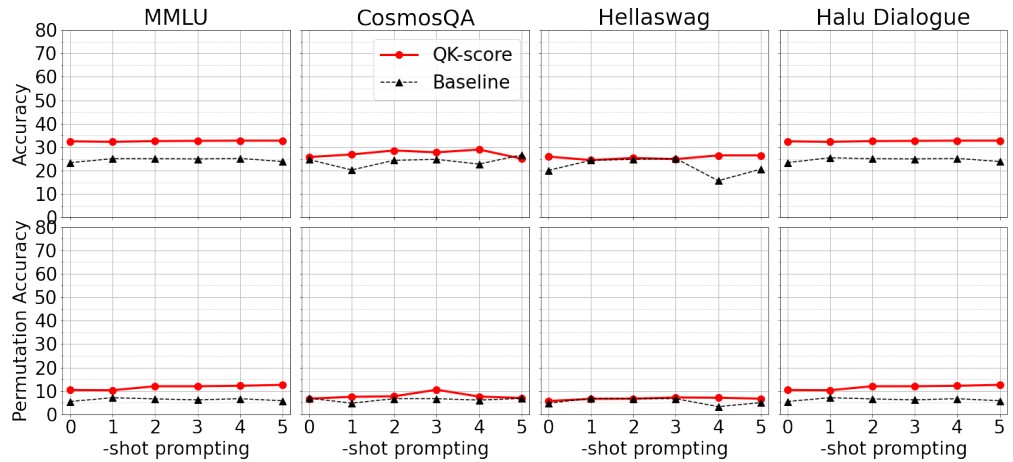

Figure 39: Comparison of different methods for Dolly V2-3B on various Q&A datasets.

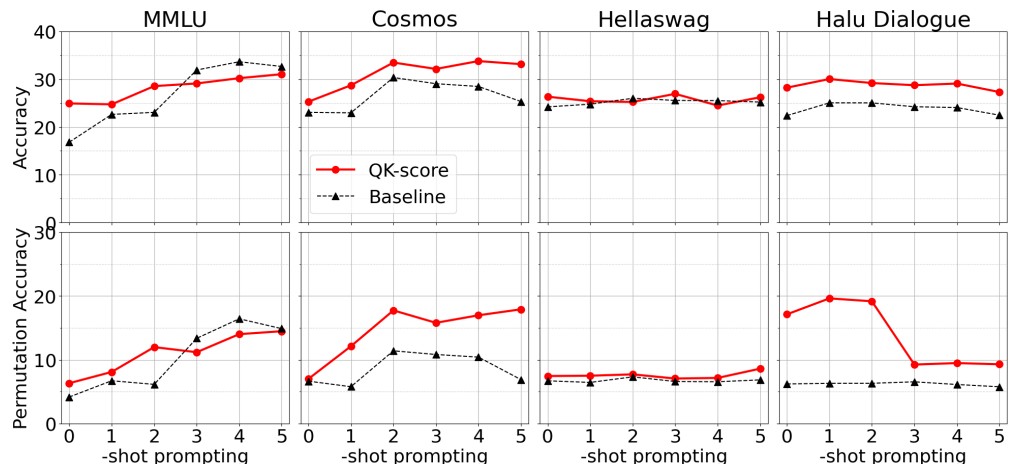

Figure 40: Comparison of different methods for Gemma-2B on various Q&A datasets.

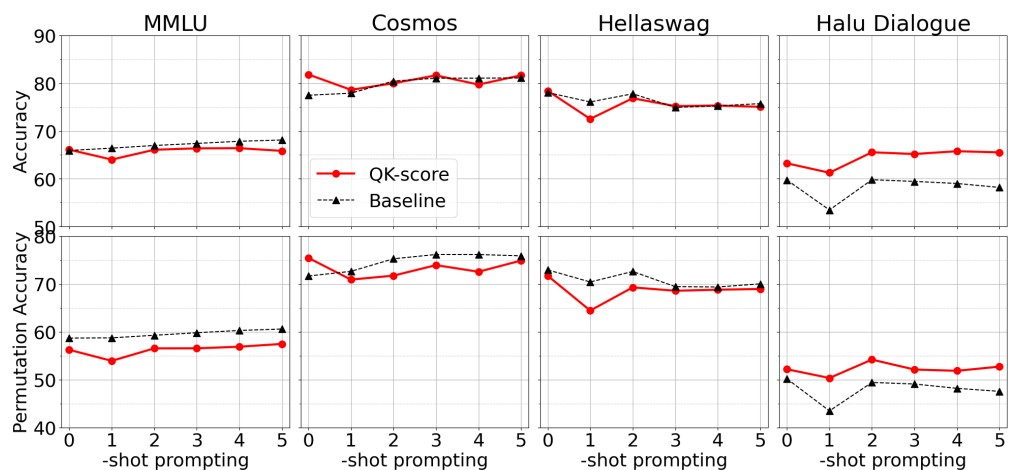

Figure 41: Comparison of different methods for Phi-3.5-mini (instruct tuned) on various Q&A datasets.

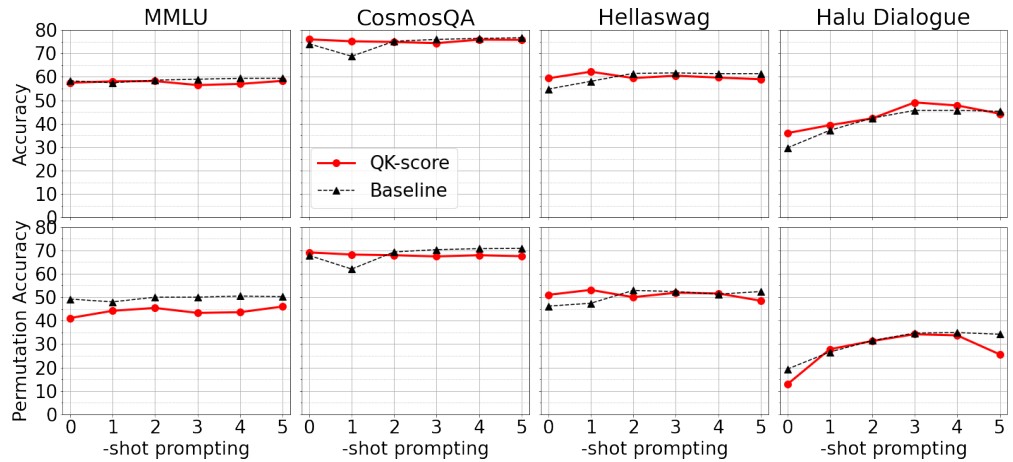

Figure 42: Comparison of different methods for Qwen-1.5B on various Q&A datasets.

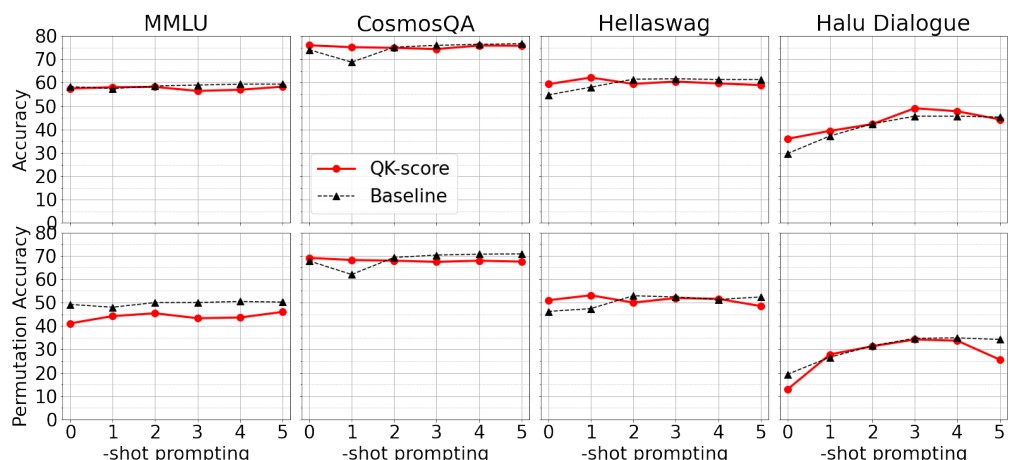

Figure 43: Comparison of different methods for Qwen-1.5B-Instruct on various Q&A datasets.

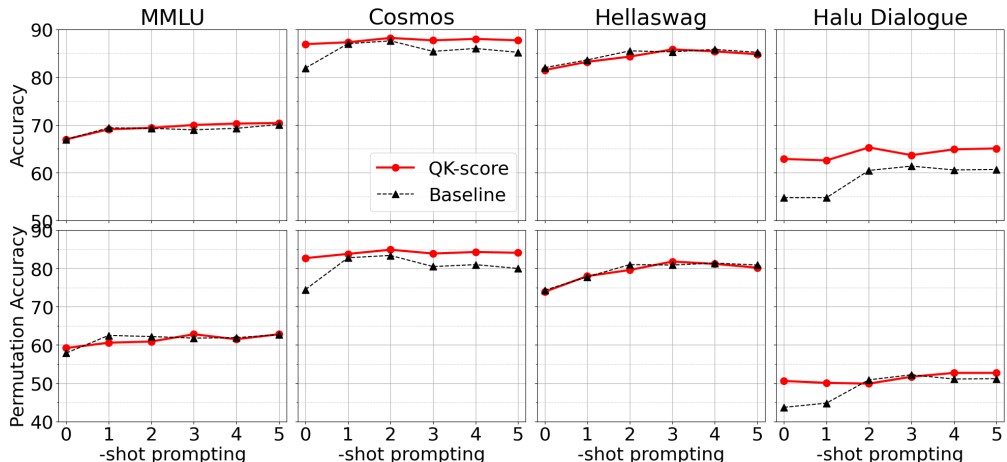

Figure 44: Comparison of different methods for Qwen-7B on various Q&A datasets.

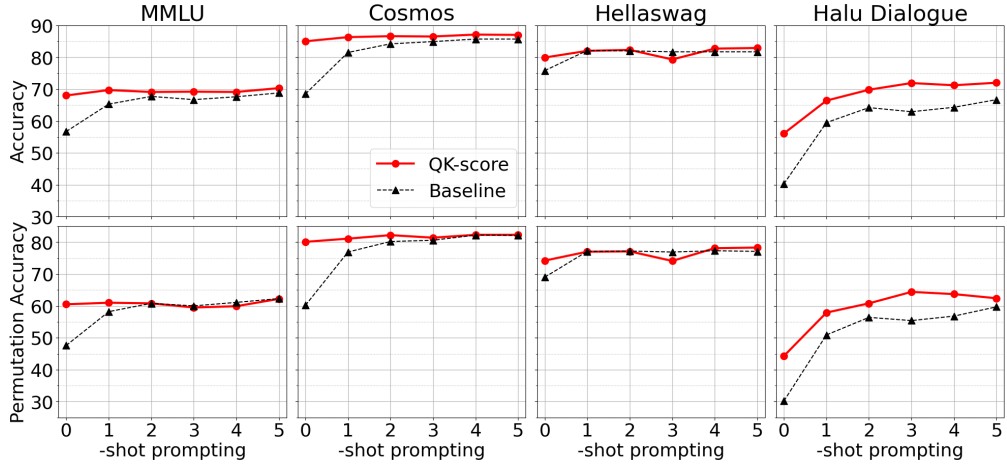

Figure 45: Comparison of different methods for Qwen-7B-Instruct on various Q&A datasets.

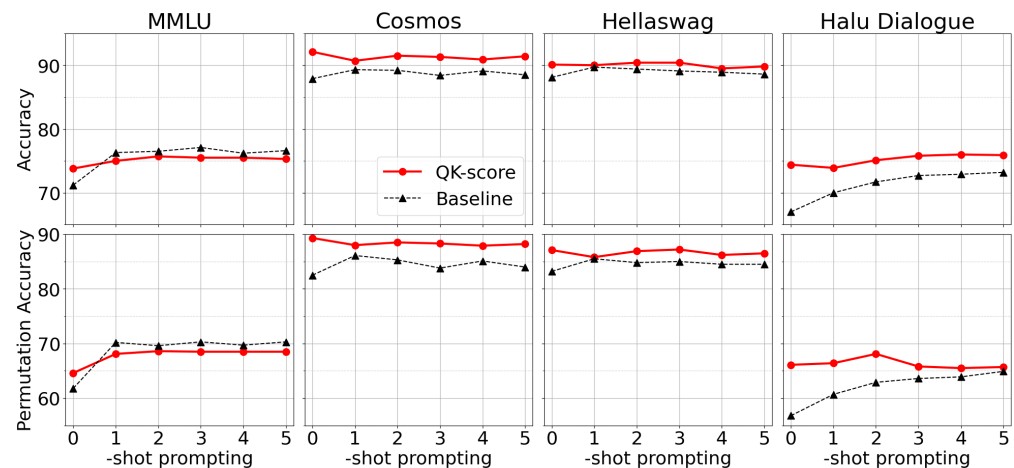

Figure 46: Comparison of different methods for Qwen-14B on various Q&A datasets.

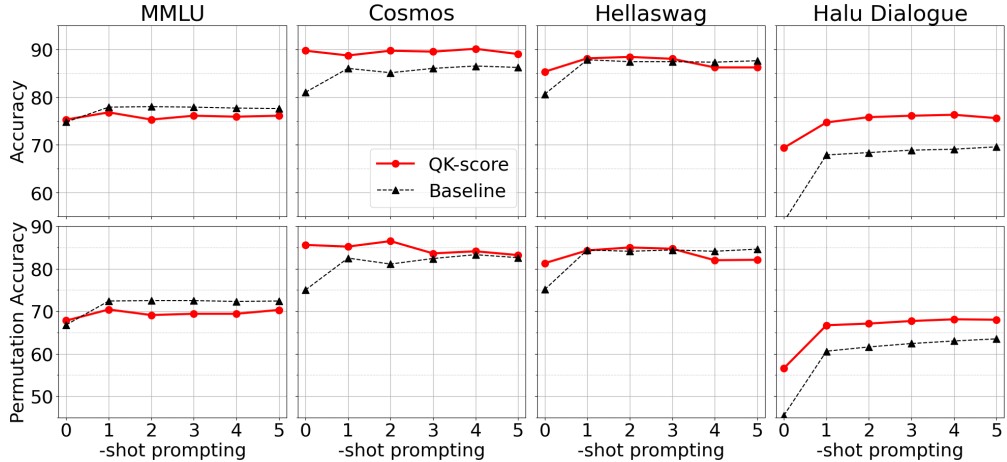

Figure 47: Comparison of different methods for Qwen-14B-Instruct on various Q&A datasets.

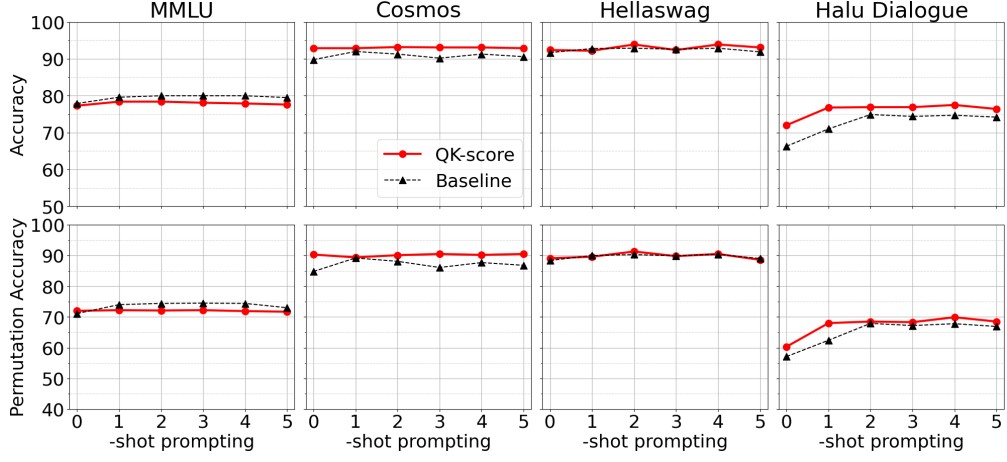

Figure 48: Comparison of different methods for Qwen-32B on various Q&A datasets.

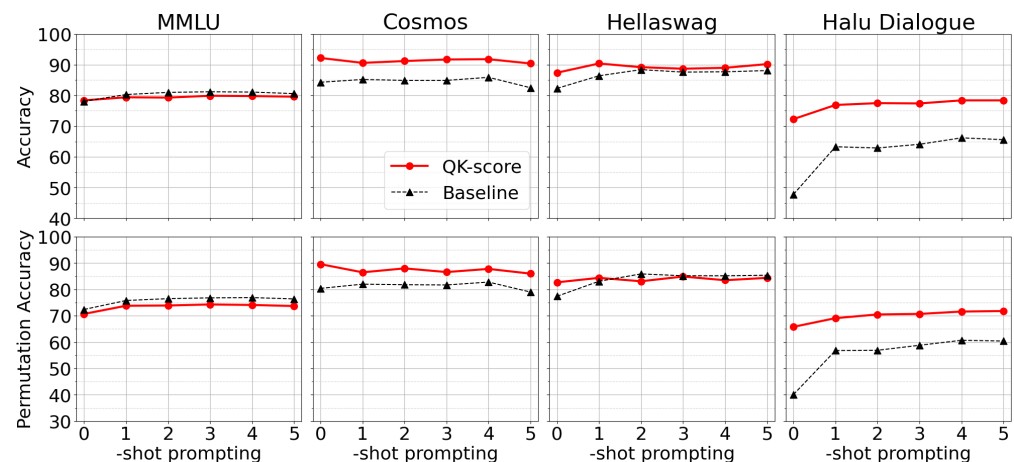

Figure 49: Comparison of different methods for Qwen-32B-Instruct on various Q&A datasets.

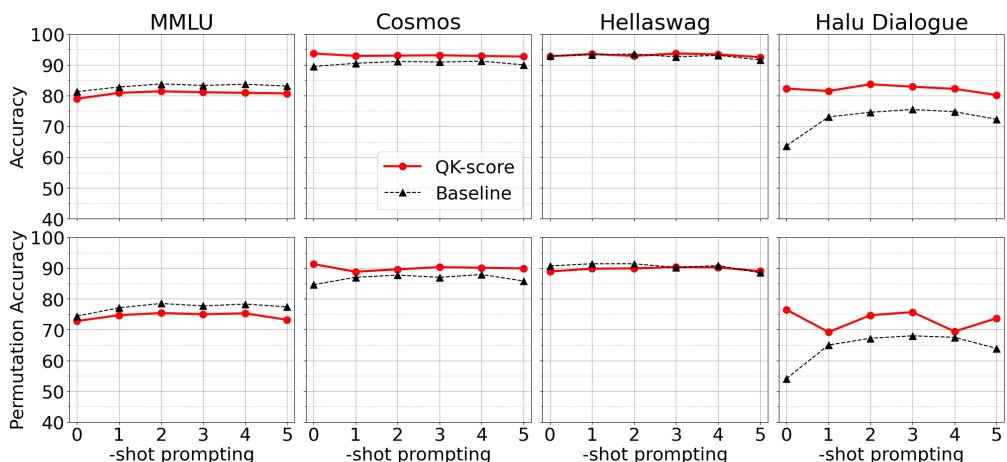

Figure 50: Comparison of different methods for Qwen-72B on various Q&A datasets.

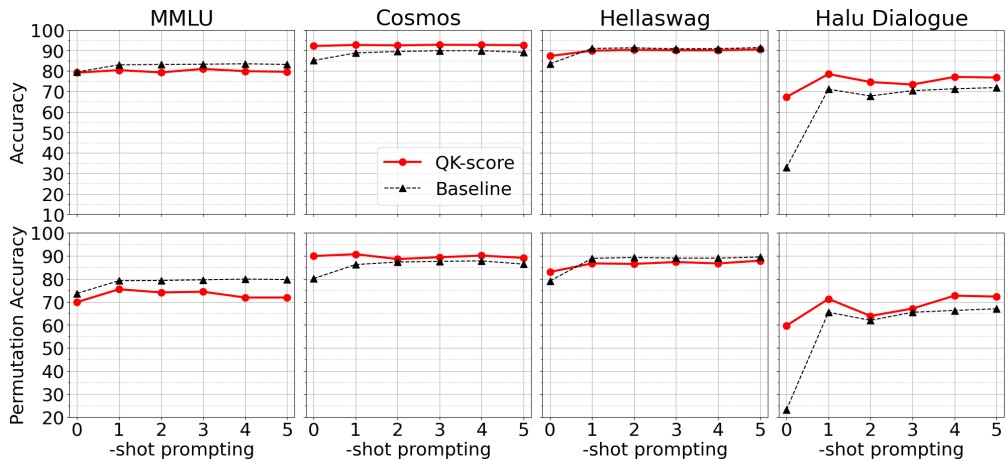

Figure 51: Comparison of different methods for Qwen-72B-Instruct on various Q&A datasets.

## N  COMPREHENSIVE RESULTS FOR EXPERIMENTS ON QWEN-2.5 AND OTHER MODEL FAMILIES

Here, we present the results of our experiments with QK-scores on four main datasets (MMLU, CosmosQA, HellaSwag, and Halu Dialogue) for models from other families. As in previous experiments, the reported metrics are Accuracy and Permutation Accuracy.

- Figures 37 and 38 contain results for DeepSeek-R1 distilled on Qwen2.5-7B and -14B.
- Figure 39 contains results for Dolly V2-3B
- Figure 40 contains results for Gemma-2B
- Figure 41 contains results for Phi-3.5-Instruct
- Figure 42 contains results for Qwen-2.5-1.5B, and Figure 43 for its instruct-tuned version
- Figure 44 contains results for Qwen-2.5-7B, and Figure 45 for its instruct-tuned version
- Figure 46 contains results for Qwen-2.5-14B, and Figure 47 for its instruct-tuned version.
- Figure 48 contains results for Qwen-2.5-32B, and Figure 49 for its instruct-tuned version.
- Figure 50 contains results for Qwen-2.5-72B, and Figure 51 for its instruct-tuned version.

We observe that the accuracy and permutation accuracy plots for the baseline and QK-scores of models around 3B in size — specifically, Dolly-v2-3B (Figure 39) and Phi-3.5-mini (Figure 41) - mostly show behavior similar to LLaMA-2 models ranging from 7B (Figure 3) to 13B (Figure 23). For these models, the QK-score is usually higher than the baseline score in zero-shot setups, and the QK-score and baseline often show convergence in few-shot setups, though at times they remain at a similar distance from each other. A similar trend is observed for the Qwen 2.5 models, with sizes of 7B and 14B, especially for instruct versions. The QK-score is also typically better than the baseline for Gemma-2B (Figure 40), although the plots for this model exhibit some unusual patterns in certain cases.

However, we observe a deviation from the general trend for both the base and instruct versions of the smallest model, Qwen 2.5-1.5B. Specifically, the QK-score and baseline scores are unusually close to each other in few-shot setups, and in several cases, the QK-score performs worse than the baseline in zero-shot setups. We hypothesize that this may be due to these models being overly fine-tuned for multiple-choice question answering (MCQA), which alters the baseline behavior in zero-shot setups. This hypothesis is supported by the fact that these models outperform LLaMA2-7B and other larger models in baseline setups.

Intrigued by these differences, we conducted an additional analysis of the behaviour of individual heads in Qwen 2.5-1.5B. We confirmed that Qwen 2.5-1.5B-base contains several heads that consistently perform well across both real and synthetic datasets, such as heads (20, 4) and (21, 11). In this regard, it remains similar to the LLaMA models. For a detailed discussion of individual head performance on the SSD dataset, see Appendix J.

## O  RESULTS FOR QK-SCORE ON FINE-TUNED QA MODELS

We have finetuned the LLaMA-2-7B on each dataset and tested how our method performs after fine-tuning. We trained LoRA adapters and merged them with the model. The results are in the Table 11. To test the models, we used the subset of data the model did not seen during the train.

## P  RESULTS FOR CLOZE PROMPTING

Cloze-style evaluation (or cloze prompting)(Robinson & Wingate, 2023) has been widely used for evaluating language models, but it has certain drawbacks. These include the "probability stealing" effect, where the correct answer's probability is spread across different surface forms(Wiegreffe et al., 2023),(Alzahrani et al., 2024). Cloze prompting is also sensitive to prompt phrasing and may lead to overfitting to training patterns. Although multi-choice prompting (MCP) addresses some of these issues, it introduces its own biases, such as position and label biases, and is sensitive to

| | | MMLU | | CosmosQA | | HaluDialogue | | HellaSwag | |
|---|---|---|---|---|---|---|---|---|---|
| | | SFT | LLaMA | SFT | LLaMA | SFT | LLaMA | SFT | LLaMA |
| 0-shot | Baseline | 0.493 | 0.267 | 0.863 | 0.311 | 0.923 | 0.211 | 0.352 | 0.265 |
| | QK | 0.488 | 0.336 | 0.840 | 0.414 | 0.905 | 0.371 | 0.393 | 0.330 |
| 1-shot | Baseline | 0.476 | 0.391 | 0.836 | 0.393 | 0.474 | 0.309 | 0.716 | 0.288 |
| | QK | 0.472 | 0.407 | 0.810 | 0.5 | 0.858 | 0.366 | 0.800 | 0.371 |
| 2-shot | Baseline | 0.477 | 0.431 | 0.643 | 0.591 | 0.823 | 0.342 | 0.795 | 0.306 |
| | QK | 0.477 | 0.421 | 0.685 | 0.615 | 0.872 | 0.406 | 0.803 | 0.404 |
| 3-shot | Baseline | 0.483 | 0.437 | 0.670 | 0.569 | 0.621 | 0.361 | 0.681 | 0.33 |
| | QK | 0.470 | 0.405 | 0.692 | 0.593 | 0.867 | 0.423 | 0.786 | 0.427 |
| 4-shot | Baseline | 0.478 | 0.441 | 0.820 | 0.579 | 0.730 | 0.345 | 0.726 | 0.361 |
| | QK | 0.470 | 0.420 | 0.802 | 0.615 | 0.877 | 0.453 | 0.789 | 0.457 |
| 5-shot | Baseline | 0.487 | 0.438 | 0.827 | 0.547 | 0.746 | 0.356 | 0.678 | 0.346 |
| | QK | 0.482 | 0.427 | 0.810 | 0.61 | 0.880 | 0.428 | 0.786 | 0.423 |

Table 11: Accuracy on supervised fine-tuned LLaMA2-7B on the same dataset the model was fine-tuned

sample order in few-shot settings. Additionally, smaller models often struggle with the required output format(Alzahrani et al., 2024), (Khatun & Brown, 2024).

As large language models (LLMs) have advanced, the format of Question Answering tasks has shifted from cloze prompting to multiple-choice formulations (Gu et al., 2024), (OpenAI, 2024), which aligns with the focus of this study. By addressing the limitations of both cloze and MCQA prompting, our method aims to provide a more reliable and insightful model evaluation.

Our method addresses several of the issues mentioned above by separating option selection from text generation within the language model. Compared to cloze and multi-choice prompting, our approach is less sensitive to answer format and wording. It also reduces common biases in MCP, such as those related to option position or the label, by disregarding the most biased attention heads. Due to the differing nature of biases, our method and cloze prompting offer complementary insights. We plan to explore how these two methods can be combined in future work.

A comparison of cloze prompting and our method is presented in Table 12. We observe that QK-score performs similarly to cloze prompting on the MMLU dataset, which requires short, knowledge-based answers, with only slight degradation in the zero-shot setting. The same holds for CosmosQA, where the model is expected to answer questions based on context. However, for datasets that assess the model's ability to continue given text snippets, QK-score significantly underperforms. This finding aligns with our understanding of QK-score as a method that separates semantic decision-making from text generation. On datasets like HellaSwag and HaluDialogue, the correct answer is often determined by the consistency of the text snippet rather than by factual accuracy or commonsense reasoning. We believe that the QK-score is more heavily influenced by the latter.

In summary, our approach achieves comparable performance while offering complementary insights into model behavior, making it a valuable alternative to traditional cloze prompting.

|         | MMLU |      | CosmosQA |      | HaluDialogue |      | HellaSwag |      |
|---------|------|------|----------|------|--------------|------|-----------|------|
|         | Cloze | QK  | Cloze    | QK   | Cloze        | QK   | Cloze     | QK   |
| 0-shot  | 0.38 | 0.35 | 0.49    | 0.46 | 0.42         | 0.40 | 0.52      | 0.38 |
| 1-shot  | 0.40 | 0.39 | 0.51    | 0.50 | 0.45         | 0.42 | 0.52      | 0.35 |
| 2-shot  | 0.39 | 0.40 | 0.48    | 0.51 | 0.46         | 0.42 | 0.53      | 0.38 |
| 3-shot  | 0.39 | 0.40 | 0.48    | 0.57 | 0.45         | 0.37 | 0.53      | 0.43 |
| 4-shot  | 0.39 | 0.39 | 0.53    | 0.54 | 0.46         | 0.39 | 0.53      | 0.43 |
| 5-shot  | 0.42 | 0.41 | 0.52    | 0.54 | 0.44         | 0.40 | 0.54      | 0.44 |

Table 12: Comparison of cloze prompting and our method with QK-Score. All experiments were ran on 4-optioned examples.

# Q  ANALYSIS OF ATTENTION MAP

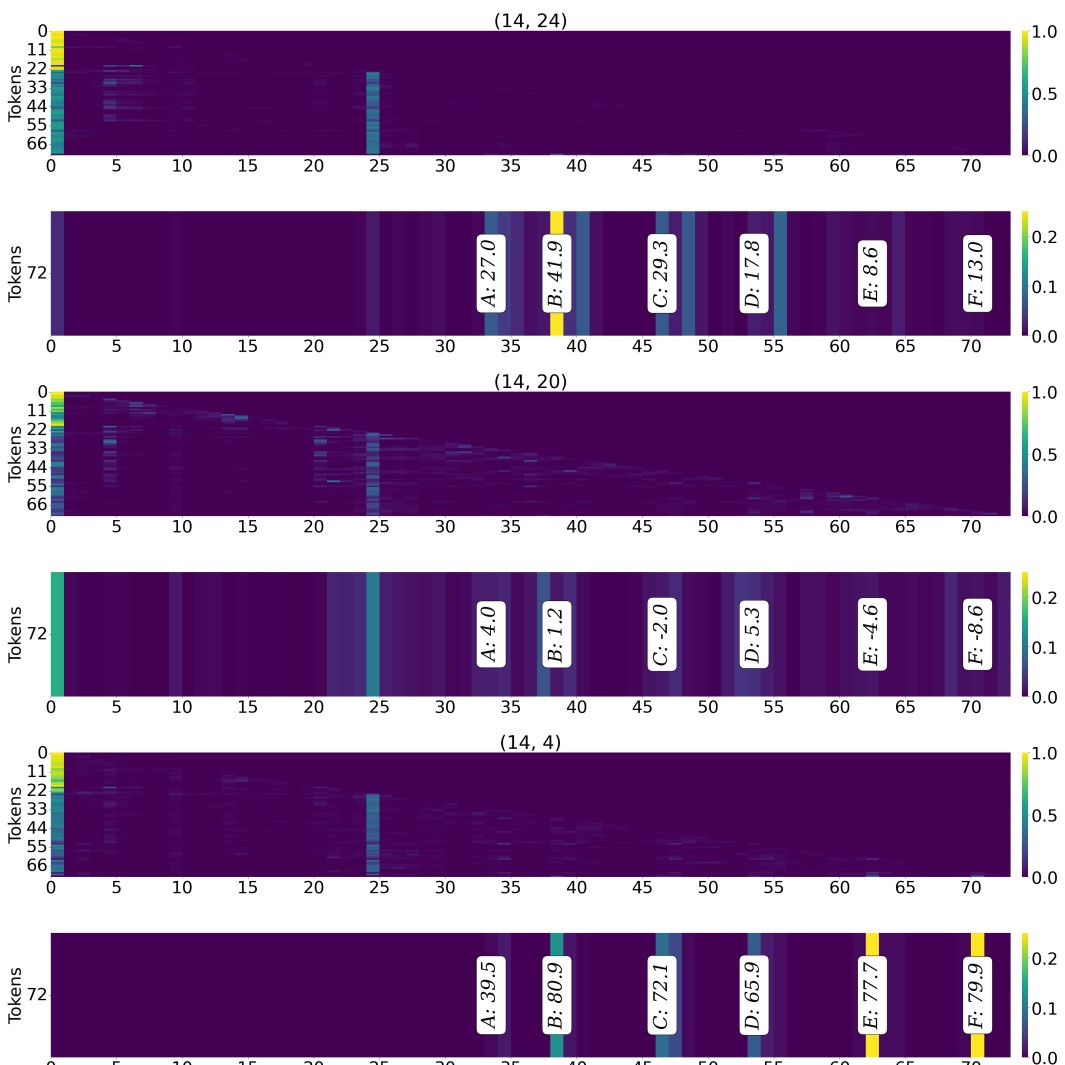

Figure 52: Attention maps of (14, 24), (14, 20) and (14,4) pairs (Layer, Head) for 0-shot setting for MMLU example: `Question: What singer appeared in the 1992 baseball film 'A League of Their Own'? \nOptions: \nA. Brandy.\nB. Madonna.\nC. Garth Brooks.\nD. Whitney Houston.\nE. I don't know.\nF. None of the above.\nAnswer:.` Second plot for each pair corresponds to the same, but scaled to the end-of-text-sequence attention map. Values in annotated cells are corresponding QK-score values. End of each option is denoted with \n symbols. 33th token is the end of A option, 38th token is the end of B option, 46th token - the end of C option, 53th token - the end of D option, 62th token - the end of E option, 70th token - the end of F option. The answer from QK-score of (14, 24) and (14, 4) is B, of (14, 20) is D. The correct answer for this example is B.

# R  HARDWARE AND RUNNING TIME INFO

The most computationally intense (per sample) part of the complete setup is the calibration when QK-scores are computed for all heads. Our main experiments with LLaMA3.1-8B (and other models with size up to 8B) were performed on 16xIntel Xeon Gold 6151 CPU @ 3.00GHz with 2x Nvidia Tesla V100 GPU acceleration and 28xIntel(R) Core(TM) i7-14700K CPU with 2x NVIDIA GeForce RTX 4090 acceleration.

Calibration on MMLU dataset (500 samples) took between 1.9 minutes for 0-shot and 2.8 minutes for 5-shot promptings.

Calibration on CosmosQA dataset (500 samples) took between 4.4 minutes for 0-shot and 6 minutes for 5-shot promptings.

Calibration on the Hellaswag dataset (500 samples) took between 4.5 minutes for 0-shot and 6.1 minutes for 5-shot promptings.

Calibration on Halu Dialogue dataset (500 samples) took between 4.6 for 0-shot and 6.5 for 5-shot promptings.

On average, evaluation of a single sample is almost 2 times faster than calibration on it.

Total running time of our main experiments for LLaMA3.1-8B (calibration and validation) took approximately 3.8/5.7/6.6/7.1 hours for MMLU/CosmosQA/Hellaswag/Halu Dialogue respectively.

Experiments with larger models ($\geq$ 14B) were performed on 8xIntel Xeon Gold 6338 CPU @ 2.00GHz with 4x Nvidia A100 80 GB GPU accelertion. For example, for the largest LLaMA3.1-70B, full computations of our main experiments ($0-, \ldots 5-$shot promptings with 500 samples for calibration and $9,500$ for evaluation in each) took $\approx 13/21/23/23$ hours for MMLU/CosmosQA/Hellaswag/Halu Dialogue respectively.

