# OpenReview forum: "Listening to the Wise Few: Query–Key Alignment Unlocks Latent Correct Answers in Large Language Models"
_ICLR.cc/2026/Conference — Submitted to ICLR 2026_

### Official Review · Reviewer_yKm9 · 2025-11-01

**Soundness:** 3
**Presentation:** 2
**Contribution:** 3
**Rating:** 4
**Confidence:** 3

**Summary:**

The paper offers a simple, principled readout (QK-score) that surfaces a real and interesting phenomenon: mid-layer “select-and-copy” heads often contain the correct MCQA choice even when the decoder output fails. The evidence spans multiple datasets, models, and ablations, which is a meaningful contribution and likely to spark follow-up work in interpretability and evaluation.

**Strengths:**

- Mechanistic evidence: by emphasizing query–key alignment (QK) rather than attention weights or hidden-state probes, the paper introduces a QK-score metric that reveals select-and-copy behavior.

- Cross settings experiments: extensive results across different models, languages, option counts, zero-/few-shot, and several option-representative tokens; includes robustness tests via option permutation, added “I don’t know/None of the above,” and zero ablation.

- Implications for evaluation: QK-score serves both as a diagnostic readout to locate layer-wise information loss and as a practical auxiliary readout for reliable multiple-choice scoring.

**Weaknesses:**

- Insufficient evidence beyond MCQA: core claims center on MCQA; extrapolation to generative QA, long-form reasoning, and tool use remains untested.

- Dependence on option-representative tokens: the method assumes stable anchors for options, automatic discovery or robust definition under freer, unstructured formats is unclear.

- Head selection coupled with the dataset: stability without a validation set or across domains is uncertain, selecting heads on the synthetic set degrades on some real tasks, indicating distribution sensitivity.

**Questions:**

1. Head selection appears post hoc. The selection criteria rely on downstream experimental results rather than an a priori, interpretability-grounded theory, which makes the method largely heuristic. Please pre-register head-selection rules and validate them out-of-domain to rule out selection bias.

2. Exposition and organization: Understanding the core contribution requires frequent back-referencing (e.g., approach in Section 4 depends on results in Section 7). I suggest restructuring so that selection principles, diagnostics, and evidence appear in a single forward-reading pipeline, to minimize cross-references.

3. Model coverage and consistency: I appreciate the many experiments across model sizes, but most results concentrate on the Llama family. In some non-Llama models, the proposed method shows no clear advantage, and coverage is uneven (e.g., some sections report Qwen but omit Phi and other models). I suggest standardizing model selection and ensuring comparable setups and metrics across families.

4. Minor: Missing reference at line 1722—“Figure ?? contains results…”.

5. Add a reproducibility or ethics statement section refer to the author guide.

---

> ### Author Response · Authors · 2025-11-25
>
> We thank the reviewer for the thoughtful and constructive feedback. Below we address each weakness and question.
>
> 1. **Evidence beyond MCQA**: we agree that our empirical focus is MCQA. Our claims in the paper are restricted to MCQA-style settings, and we will clarify this explicitly. At the same time, our method does not rely on MCQA-specific supervision: it only requires aligned QK interactions around semantically coherent spans. We will (i) restrict the scope of claims to MCQA and (ii) add a brief discussion and limitations paragraph noting generalization opportunities beyond MCQA as future work.
>
> 2. **Dependence on option-representative tokens:** we appreciate the concern. Importantly, the option-representative tokens we use are not arbitrary nor post-hoc: in causal decoder-only LMs, the final token of a span (e.g., the newline after an option) reliably aggregates the semantic content of that span through both the residual stream and attention patterns. Prior work shows that these end-of-span tokens often function as natural information sinks and carry compressed representations of the preceding text. Our choices therefore follow from the known structure of causal LMs rather than dataset-specific heuristics. As for MCQA task it is clear how to select it, using unstructured prompt provides difficulties, which we will also include in limitations.
>
> 3. **Head-selection stability and dependence on validation data:** we agree that head selection requires a validation set; however, in practice the required amount is very small (5% of the dataset suffices), and this mirrors prior work such as PriDe, which estimates selection bias using a small held-out subset. We also note that our selection procedure is already pre-registered in Section 5.3, and we will expand this section to make the rule fully explicit and deterministic. Importantly, Figure 9 shows that many select-and-copy heads are stable across datasets, indicating that head selection is not merely post-hoc fitting to a single domain. These stable heads appear consistently across models and tasks, even when selected using only minimal validation data. In the revision, we will clarify this point and highlight the empirical evidence supporting cross-dataset head robustness.
>
> 4. **Model coverage and consistency:** we thank the reviewer for this observation. All models in our current evaluation were run under the same metrics, prompts, and experimental setup; however, we recognize that this was not stated clearly enough. We will revise the text to make the shared setup explicit and easier to verify, and standardize the reporting across families.
>
> - **Minor issues:** we will correct the missing reference at line 1722. We will add the required Reproducibility and Ethics statements following the author guidelines.

---

### Official Review · Reviewer_cQyL · 2025-11-01

**Soundness:** 2
**Presentation:** 2
**Contribution:** 2
**Rating:** 2
**Confidence:** 3

**Summary:**

The authors use an approximation of the attention mechanism (select-and-copy) as the basis of a white box based intervention that improves MCQA accuracy. The main result is based on the observation the select-and-copy heads can be used in place of token outputs, which allows for an alternative means of accessing a model's "knowledge"

**Strengths:**

# originality

The select-and-copy head is a novel idea, but that is impart because it depends on the validity of the approximation (line 140) to be useful. That they find a set of benchmarks where it occurs suggest the approximation may be valid

#quality

The results are on a limited set of benchmarks and only touch on a single task MCQA. The results are reasonable, although another pass would help improve the clarity.

# clarity

I find the top level introduction unclear, the abstract and how exactly the method improves performance is not discussed much until line 178, so I was wondering how all this maps to the MCQA answers the whole time. I also found figure 3 difficult to read as all the lines overlap.
# significance

This relies on a task specific fine tuning like process, so I'm not sure how well the method will generalize. Also, the approximation is nver formally discussed so I'm not sure if it can be used for other tasks.

**Weaknesses:**

My main issue with the work is that it presents the "select-and-copy" heads as a major contribution, but does not attempt to quantify how they occur or if they are a significant phenomena in general. Additionally the Query-Key metric appears to be basically a type of fine tuning, in that it produces a non-linear transformation of the model's outputs based on a small training set. Comparing to a baseline fine tune or explaining how this is more data efficient would help

**Questions:**

Can you explain how this method works when deployed, how do you pick the answer to a MCQ without outputting a token corresponding to it? "retrieves the correct answer directly from individual attention heads." from the abstract

Was removing RoPE based on empirical observations? I'm surprised it was a significant factor

What am I supposed to gain from the visualizations of the QK scores? How should I interpret figure 6?

Generally this comes across as a weak result, I think it should be published, but it is not currently in good enough shape for ICLR.

---

> ### Author Response · Authors · 2025-12-03
>
> We thank the reviewer for the feedback. Below we address each concern one by one.
>
> - **""select-and-copy" heads as a major contribution…, but  does not  quantify how they occur or if they are a significant phenomena in general"**: Select-and-copy heads are present in all models explored in our studies (we mention it in “Key takeaways” paragraph at the end of Section 6). `Select-and-copy’ is  a behavioural pattern that multiple heads display in different degree; quantifying their number depends on a chosen threshold. For example, in LLaMA3-8B model there are 66 heads that perform better than the model itself (in the 0-shot setup) on at least 3 (out of 4) main datasets explored in the paper, and there are 7 heads that perform better than the model itself on all 4 datasets.
> Study of the exact reasons of why heads with such properties appear is beyond the scope of this paper.
>
> - **Query-Key metric appears to be basically a type of fine tuning**:  No, the proposed method is not related to the tuning of the model, because in the pipeline for QK-score we never adjust any of the model’s parameters. Perhaps a potential confusion might be caused by the wording, and in particular by the term “calibration subset”. It is needed to measure the performance of each individual attention head and identify Select-and-copy heads which are further used on the inference.  It is by no means a training set.
> And also, QK-score on its own does not perform any transformations of the models’output
>
> - **Comparing to a baseline fine tune or explaining how this is more data efficient would help**
> First, of all we would like to point out that our method is intended as an interpretability tool to expose specific circuits within language model. We do not aim at achieving best possible performance on the benchmarks; comparison with proper model fine-tuning on particular datasets is beyond the scope of this work.
> As we also show in our experiments, QK-score from head selected on synthetic data and given in-context examples (if any) - yellow line “Synthetic” on Figure 3 - demonstrates performance on par or better than the baseline (model itself).Thus, our method results in high performance (above the baseline, i.e. the model itself) even when data from the target domain is very limited.
>
> - **Was removing RoPE based on empirical observations?**
> Our method originally was based on the observation of geometric properties (alignment) of Query/Key vectors from certain tokens. When we designed it, the removal of RoPE was not the goal of its own, but it is one of the main differences between our method and attention scores. We do not measure the effect of RoPE in particular, but we report comparison of QK-score with attention (see Figures 2, 3, and 6).
>
> - **What am I supposed to gain from the visualizations of the QK scores?** Visualization of the QK-scores on Figure 6 illustrates the typical behaviour of the QK-scores from a head highly relevant to the MCQA task. It shows that the QK-score between the Query vector from the last token in the prompt and Key vectors from all previous tokens often has a distribution with a prominent peak at the line-break tokens at the end of the choice lines. It also provides a side-by-side comparison with attention from the same head to further highlight differences in their behaviour.
>
> - **How should I interpret figure 6?** Figure 6 compares (softmax-normalized) QK-score with the attention from the same head. This comparison shows that both of them peaks at the last token (linebreak) of the correct option B; but unlike attention, QK-score distribution is more concentrated and it also free from the irrelevant peaks on the first token (which in LLaMA models is artificial, i.e., does not correspond to any real part of the text).

---

### Official Review · Reviewer_8bfx · 2025-11-02

**Soundness:** 2
**Presentation:** 2
**Contribution:** 2
**Rating:** 2
**Confidence:** 4

**Summary:**

This paper investigates the inner workings of LLMs in an attempt to identify specific heads that reliably encode correct answers to multiple-choice benchmark questions.  They show that these select-and-copy heads can be identified, that they typically exist in the middle of a model, and that by extracting information from them performance on benchmarks can be improved over standard methods.  This suggests that LLMs "know more" than they seem to, or that their outputs are perhaps jumbled by later processing layers.

**Strengths:**

This is an intriguing white-box approach to directly assessing how LLMs represent knowledge and calculate outputs for MCQ tasks.  The identification of select-and-copy heads seems quite interesting, and I found it interesting to learn that LLMs correctly identify answers more often than they are given credit for.

The proposed method is fairly straightforward, although as a white-box technique, it is only applicable to open models.

The idea is natural and intriguing.

The experiments were most reasonable (with some caveats).

The ablations were helpful.

**Weaknesses:**

While I generally liked the idea of this paper, it fell down in a couple of places:

* The central results in Figure 3 are incredibly difficult to read, because all of the lines are overlapping. I think a bar chart would have been much better, or some alternative visualization that somehow magnified the difference between the various techniques.  As it is, because I cannot tell the difference between the algorithms and there are no error bars, I conclude that this method essentially does not change behavior / performance. That calls into question the entire point of the paper.

* There are a couple of times when the method genuinely seems to help.  However, given the fact that it doesn't help in 50% of the benchmark tasks, (and even then, only in the few-shot regime), it seems that more extensive empirical work is required, combined with better presentation.

* I like the idea of trying to find universal heads with a synthetic task. The heads you found didn't seem optimal, and you state that: "Comparing head-selection strategies, synthetic-data heads excel on the first two datasets but lag on the latter pair, whereas validation-selected heads deliver robust performance throughout. This suggests that tasks involving longer options, deeper reasoning, or discourse coherence require skills not captured by our simple synthetic task."

So: why didn't you design a better synthetic task?

* The identification of the SAC heads is ultimately unsatisfying -- while it's interesting to see their existence, it's not really clear how they work (and more importantly, why they are inhibited!).

Overall, I think this paper sits uncomfortably straddling two worlds: it combines elements of benchmarking and mechanistic interpretability, but does not seem to make strong contributions to either world.

**Questions:**

See weaknesses section above

---

> ### Author Response · Authors · 2025-11-25
>
> We thank the reviewer for the detailed and thoughtful feedback.
> 1. **Figure 3 readability**: Thank you for pointing this out. We agree that the current line plot becomes cluttered when many settings are overlaid. In the revision, we will rework it. We emphasize that QK-score often improves accuracy by 5–29% depending on dataset (Tables 2/5), and the revised figures will reflect this more transparently.
>
> 2. **“Does not help in 50% of tasks”**: We acknowledge the phrasing in the main text may have obscured the magnitude and consistency of the gains. While we still remain in the field of interpretability and point out that several heads can achieve comparable accuracy as the models outputs, some clarifications should be added: 1) in zero-shot, QK-score improves performance on all datasets for all LLaMA models and for most non-LLaMA models (Tables 2, N), 2) in a few-shot setting, the gap narrows because few-shot prompting already teaches the model the required MCQA format; QK remains competitive without the computational overhead of multiple demonstrations. We will expand the empirical section to better highlight where and why QK helps, and where task structure inherently limits MCQA readouts.
> 3. **Why not design a better synthetic task?** We agree that the synthetic dataset captures only the selection subproblem of MCQA (identifying the option corresponding to a target string), not long-range reasoning or discourse. Our intention was not to approximate the full complexity of real-world datasets, but to isolate a minimal, mechanistically clean setting where select-and-copy (SAC) heads should be detectable if they exist. This mirrors prior interpretability practice: start with a controlled task to reveal structure, then test transfer. In the revision we will explicitly state the purpose of the synthetic task as isolating the selection mechanism only, discuss why tasks requiring long-range reasoning cannot be captured by a synthetic MCQA minimal pair without reintroducing the full complexity of real datasets, and discuss the future work on the design of synthetic dataset.
> 4. **Why SAC heads work and why they are inhibited?** We agree that it is an important interpretability question. Our key contributions are: 1) providing direct mechanistic evidence that mid-layer SAC heads reliably identify the right option, 2) showing that later layers overwrite or gate this signal (via zero-ablation and layerwise analysis). In the revision, we will expand Section 7 with:
> - a clearer mechanistic explanation of the SAC circuit: SAC heads select the end-of-span token with the highest semantic match to the query and inject its value vector into the residual stream,
> - a connection to known phenomena where models “know” the answer internally but output the wrong format or token (e.g., cognitive dissonance, token competition).
> Our goal is not yet to fully explain why the inhibition happens, but to show where it occurs and how QK reveals the latent information. We will make this more explicit.
> 5. **“Straddling two worlds”** — interpretability vs benchmarking: We appreciate this observation. Our primary aim is mechanistic interpretability: we use MCQA not as an evaluation benchmark but as a controlled setting to expose a specific circuit (select-and-copy). We will emphasize in the introduction and conclusion that the core contribution is identifying and characterizing SAC heads as a mechanism present across many LLMs
>
> We thank the reviewer again for the insightful comments. These revisions will substantially improve clarity, empirical presentation, and the interpretability narrative of the work.

---

### Official Review · Reviewer_4sms · 2025-11-02

**Soundness:** 3
**Presentation:** 3
**Contribution:** 2
**Rating:** 4
**Confidence:** 4

**Summary:**

The goal of the work is to investigate LLM capabilities to comply with a rigid output format for better application usage - findings show that there are selection bias with multiple choice questions and that the LLMs tend to provide some undesirable formats, complicating automatic evaluation despite their answer being correct.

The paper shows how the attention heads with Query-Key scores internally identify correct answers to multiple-choice questions. They first freeze the pre-trained LMs and calculate the Q-K scores within the head selected as the best among the sampled datasets. Using the select-and-copy operation, the work shows that the middle attention layer has more answer information for such multiple choice question-answering tasks and proposes to use the selective attention head from the middle layer (e.g., 14th layer) for robust performances in those tasks. The query-key interactions show how attention mechanism interact with model input-output and the internal structures in between.

**Strengths:**

Permutation Accuracy (PA) shows meaningful results in Table 2

Figs 4 and 5 - show interesting patterns in attention heads.

SSD settings seem to be informative to see QK does great job (Table1) - but it would have been more informative if there were more comparing with other baselines?

Experimental setups are illustrated enough to replicate - e.g., prompt designs (Appendices A, D and H)

Takeaways are clear.

**Weaknesses:**

The empirical implications do not seem to be vivid and conclusive (Figs. 2 and 3)
- is the authors' implication in Fig. 2 that the QK operates better than Attention for capturing correct answers? If so, the results don't seem to be apparent to me, except for some cases content eol and label cases on Hellaswag?
- same for Fig. 3?

The proposed approach does not show robust performances in tasks with longer context, deeper reasoning, or discourse coherence

The current settings in main are too much leaning towards llama models'.
In main page, I cannot see any model families except for Llama model despite the abstract illustrating other model families including Qwen. Isn't it more complete to have them in the main page? Regarding those results in Appendix,
- Tables 3 and 4 - Qwen2.5-1.5B doesn't seem to work pretty well compared to llama models. (baselines outperform the QK-scores frequently overall)
- Fig. 21 - Qwen model do seem to work better at later layers than the middle layers unlike LLama models.
- Fig. 22 - selection bias appear extreme to zero shot settings. Llama and R1-Distill-Qwen-7B seem to show different patterns. Instruct models seem to be robust agains the selection bias than base ones. 5-shot settings mitigate those biased results both for llama and distilled reasoning models. (isn't this base performance important to show how the QK on middle layers do meaningful job on capturing the correct responses?)
- Figs.23-50 - with other models than llama family, QK scores seem relatively underperforming to the baselines; llama with higher parameters (Figs. 24, 26, 27, 29, 31), Chat, instruct models (figs. 31, 33, 34, 35, 36), models with too tiny paramters (figs. 42, 43).

Appendix G seems lack of explanation in their experimental setups

Some are arranged (Layer, Head) in Table 8, others are (Head, Layer) in Table 52 - better integrate with the one same arrangment? Now am confused with Figure 9 - is it arranged with (Layer, Head) or (Head, Layer)?

**Questions:**

See weaknesses

---

> ### Author Response · Authors · 2025-11-25
>
> We thank the reviewer for the detailed and thoughtful feedback.
> First of all, we would like to point out that, although our method can be used to improve model performance on MCQA tasks, the main goal of this work is to provide a tool to analyze inner-workings of transformer LLMs and advance their interpretability.
>
> 1. **“The proposed approach does not show robust performances in tasks with longer context, deeper reasoning, or discourse coherence”**:  Could you please clarify which part of the paper raised these concerns? The benchmarks on which we performed our experiments do not separate questions by their reasoning depth. The average length of prompt in zero-shot setup on the test subset of HaluDialogue reaches 211.67 \pm 38 tokens (by LLaMA3 tokenizer). Also, Hellaswag and Halu Dialogue datasets address the discourse coherence task (in the aspects of common sense reasoning and conversation comprehension). As the experimental results show, QK-score shows solid performance on all of them; at the same time, many of the top performing heads, identified with our method, are shared between different datasets (Figure 4; Tables 7, 8) which further supports the robustness of the method.
>
> 2. Regarding comparison with Attention on Figures 2 and 3.
> The proposed method (QK-score) is related to the attention, but unlike attention, QK-score is calculated without application of the positional embeddings (RoPE) or normalization (and this information is generally irrelevant for our purposes). The comparison with Attention on Fig. 2 and 3 shows that these differences do not significantly worsen the performance, but, on average, QK performs better than attention . One of the further advantages of QK-score over the attention is its usability in cloze prompting scenarios (due to the removal of normalization).
>
> 3. **"The current settings in main are too much leaning towards llama models'"**.  We include results for models from multiple  families in Appendices M and N (LLaMA/-2/-3, Qwen2.5, Phi3.5, Gemma, etc.). Due to the space limitations, there was a limited number of models, for which we were able to include results in the main text of the paper. We intended to show how our method performs on models with different number of parameters and compare  base/tuned models; to make these results more informative, we opted to cover models from the same family, and we choose LLaMA3/3.1 as one of the most popular open LLMs (at the time of writing). We will consider reorganizing Table 2 and expand it with results for one or two models from other families that are reported in Appendices.
>
> 4. **"Qwen2.5-1.5B doesn't seem to work pretty well compared to llama models."** Yes, on 2 out of 4 datasets, QK-score on the best head performs slightly worse than the entire model. It might be caused by the larger influence of fully-connected layers (one of the possible causes). However, we would like to point that for Qwen2.5-1.5B  the performance of the QK-score from the best head is still comparable to the performance of the entire model, and more importantly, the main goal of our work is to provide a mechanistic interpretability tool but not to improve performance of a model across all MCQA benchmarks.
>
> 5.  **“Selection bias appear extreme to zero shot settings.”** It is a well-known issue in the literature that LLMs solving MCQA task may be biased towards particular options (especially those towards the end of the list). One of the reasons why we report Permutation Accuracy in the main experiments is to address this issue; in the appendix we explore the severity of this bias. As it follows from Figure 22,  in-context examples reduce the bias; however, the study of the causes of this phenomena is beyond the scope of this work.  **“Llama and R1-Distill-Qwen-7B seem to show different patterns”** yes, indeed; we assume, this difference is a natural result of differences (in architecture or training data and procedures) between these models.
>  **“isn't this base performance important to show how the QK on middle layers do meaningful job on capturing the correct responses?”** Not exactly: the severity of the bias of the model itself (baseline) and QK-scoring from the best head aren’t directly related (bias may be a product of other parts of the model). In this experiment we analyze the distribution of the predicted answers; the main idea is that an unbiased method should yield a near uniform distribution of the predictions (i.e., its probability to answer a question correctly must be independent from the ground-truth option). The results show that QK-scores from best performing heads are usually less biased towards particular options than the predictions of the entire model. We will add these clarifications to the description of this experiment for the final version of the article.

---

> ### Author Response · Authors · 2025-11-25
>
> 6. **“...with other models than llama family, QK scores seem relatively underperforming to the baselines…”**
> We acknowledge that the QK-score from the (single) best-head on MCQA tasks performs worse than the entire model (baseline) in some cases, but the gap is usually quite small, and most of the time (especially in the zero-shot setup) QK-score performance is comparable with the baseline while it uses information only from one of the attention heads without computational overhead of the upper layers of the model. The slightly lower performance of QK-score may indicate a higher importance of information aggregation from several heads in a particular model. Also, we would like to stress that the primary aim of this work is to propose a tool for mechanistic interpretability.
>
> 7. **Appendix G seems lack of explanation in their experimental setups.** The experimental setup is given in the paragraph “Select-and-copy heads ablation” of the main text. That paragraph (and Figure 7) also contains results of the ablation experiment on LLaMA3.1-8B model, while Appendix G only provides extra results with experiments conducted under the same setup on two more models. We will include the explanation of the experimental setup to Appendix G to make it more suitable for the standalone reading.
>
> 8. Regarding the notation **“(Layer, Head) in Table 8, others are (Head, Layer) in Table 52”**. Thank you for pointing out this issue. In all tables and figures throughout the paper we always address heads in the notation ( Layer, Head).  ``(Head, Layer)'' in the caption of Figure 52 is a typo, and we will fix it. The actual heads on that Figure are listed in the correct ( Layer, Head) notation.

---

### Meta-Review · Area_Chair_ELNy · 2026-01-05

**Summary:**

Overall, the reivewers seemed to appreciate the notion of QK score, though there was significant consensus that the paper is not ready for publication, due to a combination of presentation issues and concerns with the empirical validation. I think the authors did a good job at defending the paper, but unfortunately, no discussion between reviewers and authors followed.

**Reviewer Concerns:**

I think the authors did a good job at responding to the reviewers. Unfortunately, the reviewers were not able to engage in a discussion with the authors. The following is a summary of the non-presentation-related concerns.

Reviewer 4sms seemed to mostly have issues with the experimental results and how they supported the claims. I thought the authors did a pretty good job at pushing back.

Reviewer 8bfx asked for a better synthetic task. The authors defended their choice.

Reviewer cQyL had issues with the QK score introduced by the authors, and claimed they were "basically a type of fine tuning". The authors pushed back on the interpretation, since unlike finetuning QK does not involve changing the model's parameters. I agree with the authors. More importantly, the reviewer appeared skeptical of the significance of the results and how general the phenomenon investigate in the paper is. It is unclear exactly along which axes the reviewer would have wanted to have this quantified, but nonetheless the concerns remained generally unaddressed.

Reviewer yKm9 again seemed generally skeptical of the significance / generalizability of the approach.

**Reviewer Scores:**

I believe the authors did a good job at defending their choices. Given the lack of interactions between the authors and the reviewers, it is hard to predict any score changes.

---

### Decision · Program_Chairs · 2026-01-26

Reject